# Molecular and functional profiling identifies therapeutically targetable vulnerabilities in plasmablastic lymphoma

Fabian Frontzek [1], Annette M. Staiger [2,3], Myroslav Zapukhlyak[1], Wendan Xu[1], Irina Bonzheim[4], Vanessa Borgmann[4], Philip Sander[4], Maria Joao Baptista [5], Jan-Niklas Heming[1], Philipp Berning[1], Ramona Wullenkord[1], Tabea Erdmann[1], Mathias Lutz [1], Pia Veratti[6], Sophia Ehrenfeld[7], Kirsty Wienand [8], Heike Horn[2,3], John R. Goodlad[9], Matthew R. Wilson [10], Ioannis Anagnostopoulos[11,12], Mario Lamping[12], Eva Gonzalez-Barca [13], Fina Climent[14], Antonio Salar [15], Josep Castellvi [16], Pau Abrisqueta[17], Javier Menarguez[18], Teresa Aldamiz[19], Julia Richter [20], Wolfram Klapper [20], Alexandar Tzankov [21], Stefan Dirnhofer[21], Andreas Rosenwald[11], José Luis Mate[22], Gustavo Tapia[22], Peter Lenz[23], Cornelius Miething [7,24,25], Wolfgang Hartmann [26], Björn Chapuy[8], Falko Fend [4], German Ott[3], José-Tomas Navarro [5], Michael Grau[1,27] & Georg Lenz [1,27✉]

Plasmablastic lymphoma (PBL) represents a rare and aggressive lymphoma subtype frequently associated with immunosuppression. Clinically, patients with PBL are characterized by poor outcome. The current understanding of the molecular pathogenesis is limited. A hallmark of PBL represents its plasmacytic differentiation with loss of B-cell markers and, in 60% of cases, its association with Epstein-Barr virus (EBV). Roughly 50% of PBLs harbor a *MYC* translocation. Here, we provide a comprehensive integrated genomic analysis using whole exome sequencing (WES) and genome-wide copy number determination in a large cohort of 96 primary PBL samples. We identify alterations activating the RAS-RAF, JAK-STAT, and NOTCH pathways as well as frequent high-level amplifications in *MCL1* and *IRF4*. The functional impact of these alterations is assessed using an unbiased shRNA screen in a PBL model. These analyses identify the IRF4 and JAK-STAT pathways as promising molecular targets to improve outcome of PBL patients.

A full list of author affiliations appears at the end of the paper.

Plasmablastic lymphoma (PBL) was first described in 1997 by Delecluse et al. and represents a distinct entity in the WHO classification of lymphoid tissues[1,2]. PBL is characterized by unfavorable outcome[3,4] and occurs typically at extranodal sites, predominantly the oral cavity and the gastrointestinal tract[5]. It is frequently associated with immunosuppression and patients are commonly infected by the human immunodeficiency virus (HIV)[1,6]. However, PBL also affects immunocompetent patients. PBLs are thought to arise from post germinal center B-cells that are in the transition towards plasma cell differentiation. Hence, a hallmark of PBL represents the loss of typical B-cell antigens, while plasma cell markers are strongly expressed[7].

Our current understanding of the molecular pathogenesis in PBL is very limited, but in 60% of cases, PBL cells are infected by the Epstein-Barr virus (EBV)[6,7]. Roughly 50% of PBL cases harbor MYC translocations that usually rearrange MYC to the heavy chain immunoglobulin locus suggesting an essential role of MYC in the biology of PBL. Recently, a small study using targeted sequencing identified PRDM1, STAT3, BRAF, and EP300 mutations in primary PBL samples[8], whereas another analysis focused on HIV-associated PBL that revealed recurrent mutations of the JAK-STAT pathway[9]. However, due to the small sample size respectively the focus on HIV positive PBL, the landscape of genetic aberrations in PBL, and the exact functional role of these abnormalities in the molecular pathogenesis of PBL remain largely unknown. To improve the outcome of PBL patients a significantly better understanding of the biology is warranted.

In this work, we perform a comprehensive genomic analysis describing the mutational landscape of the entire exome and genome-wide somatic copy number alterations (SCNAs) in PBL. We reveal recurrent genetic alterations affecting the RAS-RAF, JAK-STAT, MCL1, IRF4, and NOTCH pathways as PBL defining molecular markers and potential therapeutic targets.

## Results

### Histology, immunohistochemistry, and fluorescence in situ hybridization.
We collected 96 formalin-fixed and paraffin-embedded (FFPE) primary PBL samples (Supplementary Fig. 1a, b, c). For all selected cases the diagnostic criteria according to the WHO classification of 2017 were fulfilled and confirmed in a central pathology review by a panel of expert hematopathologists.

As expected, 86% (82/95) of PBLs in our cohort did not express the B-cell antigen CD20, while the remaining 14% (13/95) showed weak and inconsistent expression. Of all cases, 57% (55/96) exhibited latently EBV-infected tumor cells, while 33% (17/52) of patients with available HIV infection status were HIV positive. EBV positive PBL cases were not associated with special morphologic PBL subtypes compared to EBV negative cases (data not shown).

Sixty-eight cases were assembled on a tissue microarray to uniformly perform immunohistochemical staining for selected markers (Supplementary Fig. 1a and Supplementary Data 1). All cases exhibited strong reactivity of the plasma cell marker IRF4 (68/68). The proliferation index Ki-67 was high with a median value of 80% (range: 30–100%). To determine the frequency of MYC translocations, we performed fluorescence in situ hybridization (FISH) using an MYC dual color break-apart rearrangement probe (BAP) in 57 evaluable cases (Supplementary Fig. 2a) and an MYC-IgH fusion probe (FP) in 63 cases (Supplementary Fig. 2b). In total, 47% (28/60) of cases harbored an MYC translocation, determined by positivity for BAP and/or FP. Since 35% of MYC-BAP positive cases (8/23) were negative for MYC-FP, about one third of cases translocated MYC to a non-IgH partner (Supplementary Data 1). The MYC translocation status

was associated with a significantly higher Ki-67 index ($p = 8 \times 10^{-4}$, one-tailed two-sample t-test).

**Mutational landscape of PBL.** To characterize the mutational landscape of PBL, we performed WES in 85 primary PBL cases (Supplementary Fig. 1a, b, c). An overview of our analysis pipeline is provided in Supplementary Fig. 3. We obtained a median effective coverage for all samples and all exons of 80 reads (Supplementary Fig. 4). After variant discovery and filtering (for details see methods), we called somatic mutations with an average of 3.48 variants per Mb and sample, placing PBL among other cancer entities with moderate to high tumor mutational burden (TMB) (Supplementary Data 2 and Supplementary Fig. 5). After detecting somatic mutations in individual samples, we aimed to identify recurrently mutated putative cancer candidate genes (CCGs) and applied to this end the MutSig2CV algorithm (for details see Methods). This analysis revealed 15 CCGs ($q < 0.1$, cohort frequency ≥5%; Fig. 1a, Supplementary Fig. 6, and Supplementary Data 3, 4).

Recurrent mutations affected the RAS-RAF pathway. The oncogene NRAS was the most frequently mutated gene in 31% of cases (Fig. 1a). All NRAS variants were missense mutations and—except one—occurred exclusively at the known hotspot residues p.G12, p.G13, and p.Q61 (Fig. 1b). Mutations of KRAS and HRAS occurred in 11% and 2% of cases, respectively (Fig. 1a and Supplementary Data 4). NRAS and KRAS mutations were mutually exclusive. BRAF mutations were found in 6% of cases and were all located in the kinase protein domain (Fig. 1b, c). Altogether 47% (40/85) of PBL samples displayed a RAS or BRAF mutation (Fig. 1d). The majority of detected mutations were clonal, indicating their role as initial oncogenic driver genes (Fig. 1a).

We additionally detected recurrent mutations activating the JAK-STAT pathway. Most frequently, we found STAT3 mutations in 25% of PBL cases (Fig. 1a). Virtually all STAT3 mutations clustered in the SH2 domain that is essential for dimerization and activation of STAT3[10], including the p.D661 and p.Y640 residues that were affected in 33% (7/21) and 24% (5/21) of cases, respectively (Fig. 1b, c). Roughly 50% of the detected STAT3 mutations were clonal underlining their role as CCGs (Fig. 1a). Notably, STAT3 mutations occurred in only 10% (3/31) of HIV negative patients, but in 47% (7/15) of HIV infected individuals ($p = 0.003$, $q = 0.043$, Wilcoxon test; Supplementary Data 5), suggesting a pathogenetic role of STAT3 mutations especially in HIV-associated PBL. JAK1-3 missense mutations occurred in 5% (4/85) of cases, while genes encoding the JAK-STAT inhibiting proteins SOCS1 and/ or SOCS3 were mutated in 12% (10/85) of samples at multiple sites spanning the entire open reading frame suggesting loss of function. These alterations did not occur in a mutually exclusive fashion, as 29% of PBLs with STAT3 mutation (6/21) harbored concomitant alterations of JAK1-3, SOCS1/3, or PIAS3. Altogether 35% of PBL cases (30/85) harbored mutations affecting the JAK-STAT pathway (Fig. 1d).

In 26% (22/85) of samples mutations affecting genes encoding for components of the NOTCH signaling pathway were detectable, including mutations in SPEN (8%), NOTCH1 (7%), NOTCH4 (6%), DTX1 (3%), NOTCH2 (1%), and NOTCH3 (1%) indicating a role of NOTCH signaling in the molecular pathogenesis of PBL (Supplementary Data 4 and Fig. 1d).

In addition, we found several genetic alterations inactivating tumor suppressor genes. TP53 mutations were frequent and occurred in 14% of PBL cases (Fig. 1a). These mutations clustered particularly in the DNA binding domain (exons 5 to 8) (Supplementary Fig. 7a). Interestingly, only 27% of TP53 mutations were clonal suggesting that the majority of TP53

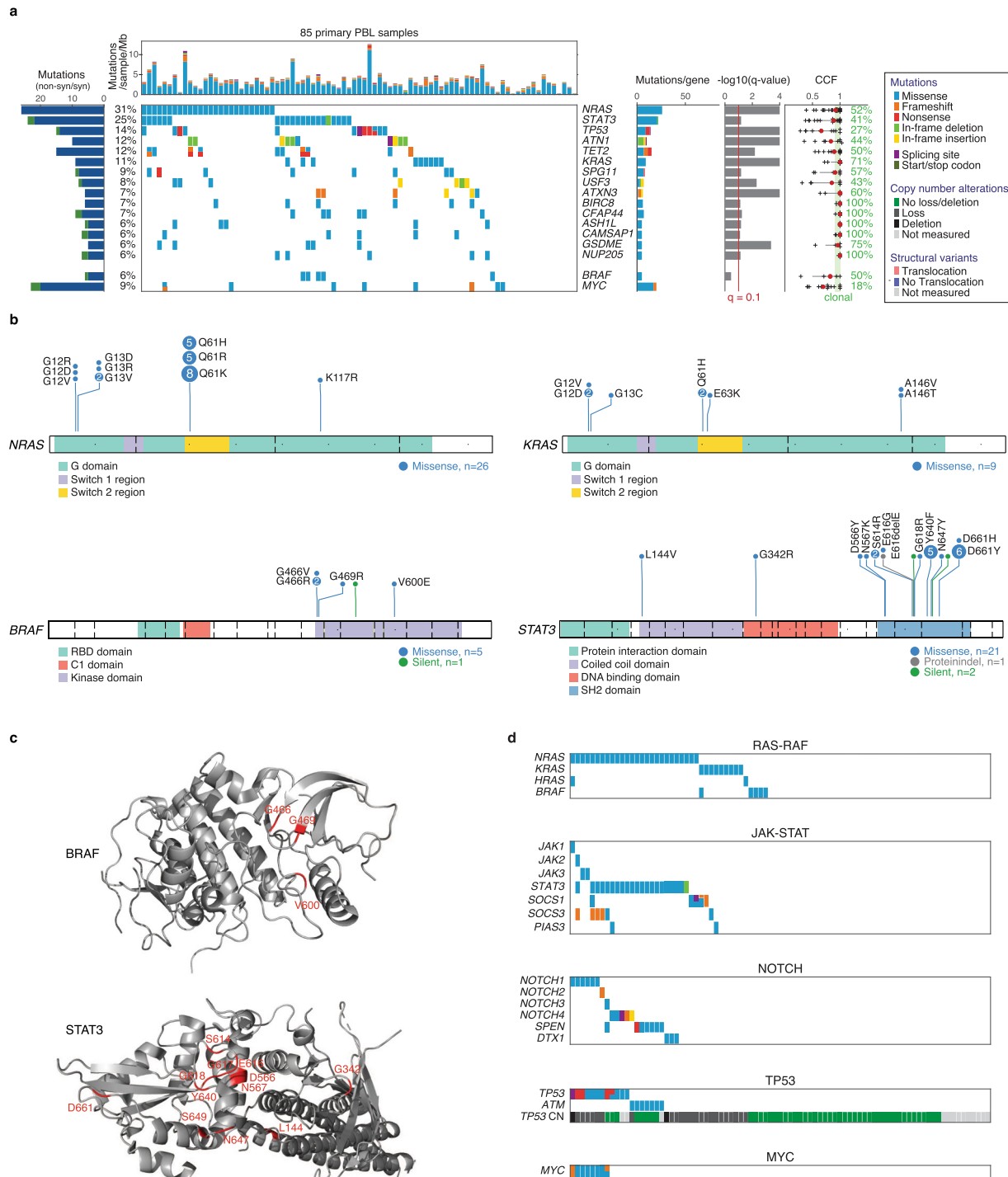

**Fig. 1 Landscape of somatic mutations in PBL determined by WES. a** All called non-synonymous mutations in significant genes according to MutSig2CV v3.11 ($q_{M2CV}$ < 0.1, cohort frequency ≥5%) are color-coded and shown for each PBL sample per column, ranked by cohort frequency (see Supplementary Data 3 and 4 for all results). Samples are ordered by waterfall sorting based on binary gene mutation status. The bar graph on the left shows the ratio of non-synonymous (blue) and synonymous (green) mutations per gene. At the top, the TMB per sample (mutations/sample/Mb) is depicted. On the right, occurring types of mutation and *q* values (M2CV) are shown per gene. For each gene, the CCF (fraction of cancer cells having a mutation in at least one allele) was estimated for samples with corresponding copy number measurement (median in red). Clonality was assumed for CCF ≥0.9. The percentage of samples with clonal mutations is indicated per gene. **b** The distribution of detected mutations on protein level for the selected CCGs *NRAS* (NM_002524), *KRAS* (NM_033360), *BRAF* (NM_004333), and *STAT3* (NM_139276). Exon boundaries are indicated using dashed lines. **c** Spatial clustering of mutations within the protein structures of BRAF (PDB 6nyb)[95] and STAT3 (PDB 6njs)[96]. **d** Co-occurrence of mutations belonging to selected biological pathways. For each analyzed pathway, samples are presented in their corresponding waterfall sort order by binary gene mutation status. Copy number (CN) and translocation status (SV) are depicted for *TP53* and *MYC*, respectively.

mutations represent a later pathogenic event (Fig. 1a). TP53 mutations occurred in 6% of EBV positive lymphoma cases (3/47), while EBV negative PBLs harbored TP53 mutations in 24% (9/38) ($p = 0.011$, $q = 0.193$, Wilcoxon test; Supplementary Data 5). In 70% of samples with TP53 mutation and available copy number status (7/10) we detected concomitant loss of the alternate allele leading to biallelic inactivation (Fig. 1d). The TP53-stabilizing gene ATM harbored missense and frameshift mutations in 8% of cases, which were mutually exclusive to TP53 mutations. We also identified mutations in the tumor suppressor KLHL6 in 8% (Supplementary Fig. 7b). As previously reported in other lymphoma subtypes, in 86% (6/7) of PBL cases with identified KLHL6 mutations the mutations clustered in the hotspot domain BTB, possibly leading to a dissociation of KLHL6 and cullin3 that form a functional cullin-RING ubiquitin ligase. KLHL6 mutations have been described to disrupt this ligase function and to contribute to the growth of diffuse large B-cell lymphoma (DLBCL) cells in vitro and in vivo[11–13].

Several genes encoding epigenetic regulators were identified among the potential CCGs such as the tumor suppressor TET2 that harbored inactivating missense, nonsense, or frameshift mutations in 12% (Fig. 1a and Supplementary Fig. 7c). The genes encoding the methyltransferases KMT2A and KMT2D exhibited mutations in 8% and 6%, respectively (Supplementary Data 4).

Finally, we identified MYC mutations in 9% of cases that were predominantly subclonal (Fig. 1a and Supplementary Fig. 7d). Of note, 27% (7/26) of MYC translocated cases were concomitantly mutated, while untranslocated cases did not display a mutation (0/32) ($p = 0.001$, $q = 0.017$, Wilcoxon test, Supplementary Data 5 and Fig. 1d) suggesting that MYC mutations originate from somatic hypermutation[14].

To test the specificity of variant filtering, we systematically compared the results of PBL cases with matched normal ($n = 22$) vs. PBL cases without available paired normal tissue ($n = 63$) (Supplementary Figs. 8 and 9). The median TMB per megabase was 1.87 for PBL cases with paired normal vs. 3.58 for PBL cases without. However, comparing top-mutated genes (frequency ≥10%) and adjusting for multiple hypothesis testing, MUC4 was the only gene being significantly more frequently mutated in the cohort of PBL samples without matched normal ($p = 0.0004$, $q = 0.0194$, Wilcoxon test; Supplementary Data 5). This indicates that our filtering strategy allows for sufficiently specific calling of somatic mutations in top-mutated genes even for patient samples without available paired normal tissue.

We next validated our results using amplicon-based deep targeted resequencing for the selected CCGs NRAS, KRAS, BRAF, STAT3, TP53, and TET2 in 54 primary PBL samples of our study cohort for which sufficient DNA was available. Forty-eight of 49 mutations (98%) called by WES were independently confirmed. For nine cases, the wildtype status determined by WES was confirmed. Eleven variants detected by targeted resequencing had also been discovered by WES but subsequently filtered out as germline variants or as an artifact. Ten further identified variants had an allele frequency (AF) below our defined threshold of 10%. A total of nine mutations (three mutations with AF 10–16% and six mutations with a higher AF) were additionally revealed by targeted resequencing suggesting that the mutational cohort frequencies might be slightly underestimated due to a lower effective coverage for single genomic sites using WES (Supplementary Data 6). Specifically, when combining results of targeted resequencing and WES, STAT3 mutation frequency increased from 25% (21/85) to 28% (24/85), for TP53 from 14% (12/85) to 18% (15/85), and for KRAS from 11% (9/85) to 12% (10/85). In contrast, no additional NRAS, BRAF, or TET2 mutations were discovered using targeted resequencing. Overall, only 5.2% of all covered genomic regions showed less than 20 reads of effective coverage in WES (Supplementary Fig. 4).

**Recurrent somatic copy number alterations in PBL**. Using the Oncoscan platform, we next analyzed 82 PBL samples for copy number alterations and identified SCNAs using GISTIC v2.0.23[15] (Fig. 2 and Supplementary Data 7, 8). Of those, 16% (13/82) displayed polyploidy as determined by ASCAT[16]. Arm-level amplifications affected particularly chromosomes 1q, 7p, and 7q and were detectable in 42%, 32%, and 33% of cases, respectively (Fig. 2a). Additionally, we found various focal amplifications with 1q23.1 being the most specific identified in 61% of samples. This region contained only the genes encoding the Fc receptor-like proteins (FCRL1-5) and CD5L ($q_{G2.0} = 3.7 \times 10^{-39}$). FCRL1-5 are known to regulate B-cell development and differentiation[17]. A wider amplification in 1q21.3 affected 60 genes and occurred in 52% of samples ($q_{G2.0} = 3 \times 10^{-4}$). Within this aberration, we identified the antiapoptotic gene MCL1 that represents a drug-gable molecular target[18]. 6p25.3 was focally amplified in 29% of PBL cases and comprised only four genes including IRF4 ($q_{G2.0} = 6.4 \times 10^{-6}$). The corresponding minimal common region (MCR) and the extended peak interval are shown in Fig. 2b. In 32%, we found an amplification of 8q24.13 containing TRIB1 ($q_{G2.0} = 6.7 \times 10^{-6}$) that is also amplified in acute myeloid leukemia (AML) and is known to induce MEK1/ ERK signaling[19]. An amplification of 17q22 comprising the oncogene MSI2 was detectable in 21% of cases. MSI2 has been reported to be overexpressed in AML and to contribute to poor survival[20]. Finally, we detected recurrent amplifications of 11q23.3 affecting KMT2A that was also recurrently altered by mutations as described above.

Deletions occurred generally less frequently than amplifications (Fig. 2c). Recurrent arm-level deletions affected chromosomes 13q, 17p, 18p, and 18q and were detectable in 17%, 26%, 18%, and 16% of cases, respectively.

Focusing on focal genetic lesions, we identified deletions at 1p22.1 affecting the gene encoding the potential tumor suppressor RPL5 ($q_{G2.0} = 3 \times 10^{-3}$) in 24% of cases. Besides, we detected focal deletion of 4q35.2 and 6q26 in 26% and 25% of cases involving the tumor suppressor genes FAT1 ($q_{G2.0} = 3 \times 10^{-3}$) and PRKN ($q_{G2.0} = 2 \times 10^{-3}$), respectively.

**IPI and EBV status dictate survival in patients with PBL**. For 49 PBL patients, we were able to obtain corresponding clinical data (Supplementary Data 1). As summarized in Supplementary Table 1, 69% of patients were male and their median age at diagnosis was 62 years. Nine patients (22%) showed a high-risk International Prognostic Index (IPI) while 19 patients (46%) belonged to the intermediate and 13 patients (32%) to the low IPI risk group, respectively. With a median follow-up of 21.5 months, patients showed a 2-year overall survival (OS) of 61% (Fig. 3a). 82% of patients were treated with a CHOP-like regimen. As expected, the IPI provided a risk stratification with the high-risk group being characterized by a poor 2-year OS of only 11% (Fig. 3b; $p = 6.1 \times 10^{-6}$ for IPI low/intermediate vs. IPI high, log rank test).

Next, we investigated whether selected genetic alterations and/or clinical parameters (Supplementary Data 9) influence the prognosis of affected patients. We examined the lymphoma-specific survival (LSS) in order to exclude potential bias of disease-unrelated death causes and we focused on patients who received CHOP-like chemotherapy to ensure that unfavorable outcome was not simply due to inefficient treatment approaches. False discovery rates (FDR) were calculated for preselected biological conditions. Patients with EBV negative PBL showed a significantly inferior LSS compared to patients with EBV positive disease ($p = 0.002$, $q = 0.013$, log rank test; Fig. 3c). As described above, we detected that negative EBV status correlates with TP53 mutation. Despite a very low number of cases, patients with TP53

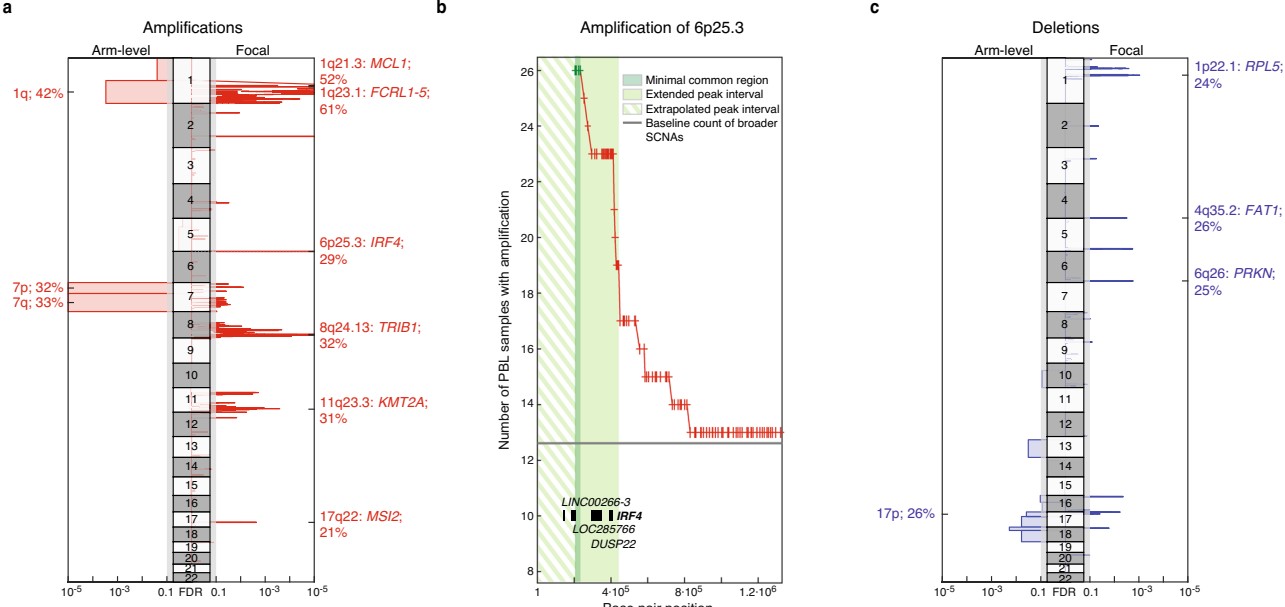

**Fig. 2 Somatic copy number alterations in primary PBL cases.** GISTIC v2.0.23 defined copy number **a** amplifications (red) are visualized for each chromosome, arm-level alterations on the left, and focal lesions on the right. Insignificant lesions ($q_{G2.0} > 0.1$) are shaded in gray. Selected potential driver genes within significant focal lesions are highlighted with corresponding cohort frequencies. **b** Pileup plot depicting the number of primary PBL samples with copy number amplifications at the beginning of chromosome 6 (cytoband 6p25.3). The minimal common region (MCR) determined by GISTIC and the robustly extended peak region is highlighted in green. Contained genes are labeled. The MCR starts with the first available SNP probe (marked by red crosses) on the chromosomal arm while only noncoding genes LOC285766 and LINC00266-3 are located left in the extrapolated extended peak region (striped). **c** Copy number deletions (blue) are correspondingly shown for each chromosome. See Supplementary Data 7 and 8 for all results.

mutated PBL showed a significantly inferior outcome compared to patients with wild-type *TP53* ($p = 0.035$, $q = 0.140$, log rank test; Fig. 3d). Patients harboring *NRAS* mutations also showed a trend towards unfavorable LSS ($p = 0.062$, $q = 0.187$, log rank test; Fig. 3e). *MYC* translocation, *MYC* expression, and HIV infection were not associated with LSS.

To investigate whether our approach selecting specific genetic alterations and/or clinical parameters may have missed alterations dictating survival, we performed an unbiased approach using a Cox-regression model to calculate $p$ values and hazard ratios describing the clinical impact for all detected significant genetic lesions ($q_{M2CV/G2.0} < 0.1$) and clinical parameters (Fig. 3f). Indeed, this analysis did not yield any additional genetic alterations or clinical parameters that influenced survival significantly in the context of multiple hypothesis testing but confirmed our findings (Supplementary Data 9).

**Genetic heterogeneity in PBL.** To further understand the genetic heterogeneity in PBL, we systematically compared the profiles of recurrent mutations and SCNAs ($q < 0.1$, cohort frequency ≥5%) for biologically defined subgroups. To this end we investigated the following groups: HIV positive ($n = 17$) vs. HIV negative patients ($n = 35$), EBV positive ($n = 55$) vs. EBV negative disease ($n = 41$), HIV positive patients with EBV positive PBL (HIV + /EBV + , $n = 15$) vs. HIV positive patients with EBV negative PBL (HIV + /EBV−, $n = 2$), HIV negative patients with EBV positive PBL (HIV−/EBV + , $n = 17$) vs. HIV negative patients with EBV negative PBL (HIV−/EBV−, $n = 18$), *MYC* translocated ($n = 28$) vs. *MYC* untranslocated PBL ($n = 32$), CD20 negative ($n = 82$) vs. weakly CD20 positive PBL ($n = 13$), PBL arising in the oral cavity/pharynx ($n = 18$) vs. PBL arising elsewhere ($n = 36$), patients with ($n = 32$) vs. without immunosuppression ($n = 29$), patients with high IPI ($n = 9$) vs. low/intermediate IPI ($n = 32$),

patients with LSS of less than 12 months ($n = 8$) vs. patients with LSS of more than 24 months ($n = 15$). We applied one-tailed Wilcoxon rank-sum tests on available data for each selected genetic lesion (Supplementary Data 5).

As described above, *STAT3* mutations were significantly more frequent in HIV positive compared to HIV negative patients while recurrent amplifications and deletions did not significantly differ. In patients with EBV negative PBL, we detected focal deletions of 1p22.1 (46% vs. 11%, $p = 0.0007$, $q = 0.0138$) and arm-level deletions of 13q (27% vs. 7%, $p = 0.0045$, $q = 0.0423$) as characteristic genetic alterations in comparison to EBV positive disease. We did not detect any significant differences in the incidence of specific mutations. Possibly in part due to limited case numbers, no significant differences were detectable in the subgroup comparisons of HIV−/EBV + vs. HIV−/EBV− and HIV + /EBV + vs. HIV + /EBV−.

In PBLs harboring an *MYC* translocation compared to non-translocated cases, we detected a pattern of several focal amplifications: 1q43 (58% vs. 21%, $p = 0.0122$, $q = 0.0754$), 2q31.3 (46% vs. 14%, $p = 0.0019$, $q = 0.0603$), 11q23.3 (50% vs. 18%, $p = 0.0046$, $q = 0.0619$), 11q25 (46% vs. 14%, $p = 0.0122$, $q = 0.0754$), 12p11.22 (42% vs. 14%, $p = 0.0060$, $q = 0.0619$). Analyzing the mutational profiles, *MYC* was the only gene being significantly more frequently mutated as described.

Comparing classically CD20 negative PBLs to weakly CD20 positive cases did not reveal any differences. In our cohort, we identified 18 PBL cases with involvement of the oral cavity while in 36 patients the oral cavity was not affected (Supplementary Data 1). Interestingly, mutations of *CFAP44* occurred significantly more frequently in PBL arising in the oral cavity (27% vs. 0%, $p = 0.0016$, $q = 0.0268$) while the distribution of SCNAs did not differ.

Overall, 52% of PBL patients (32/61) suffered from immuno-deficiency comprising HIV infection but also autoimmune

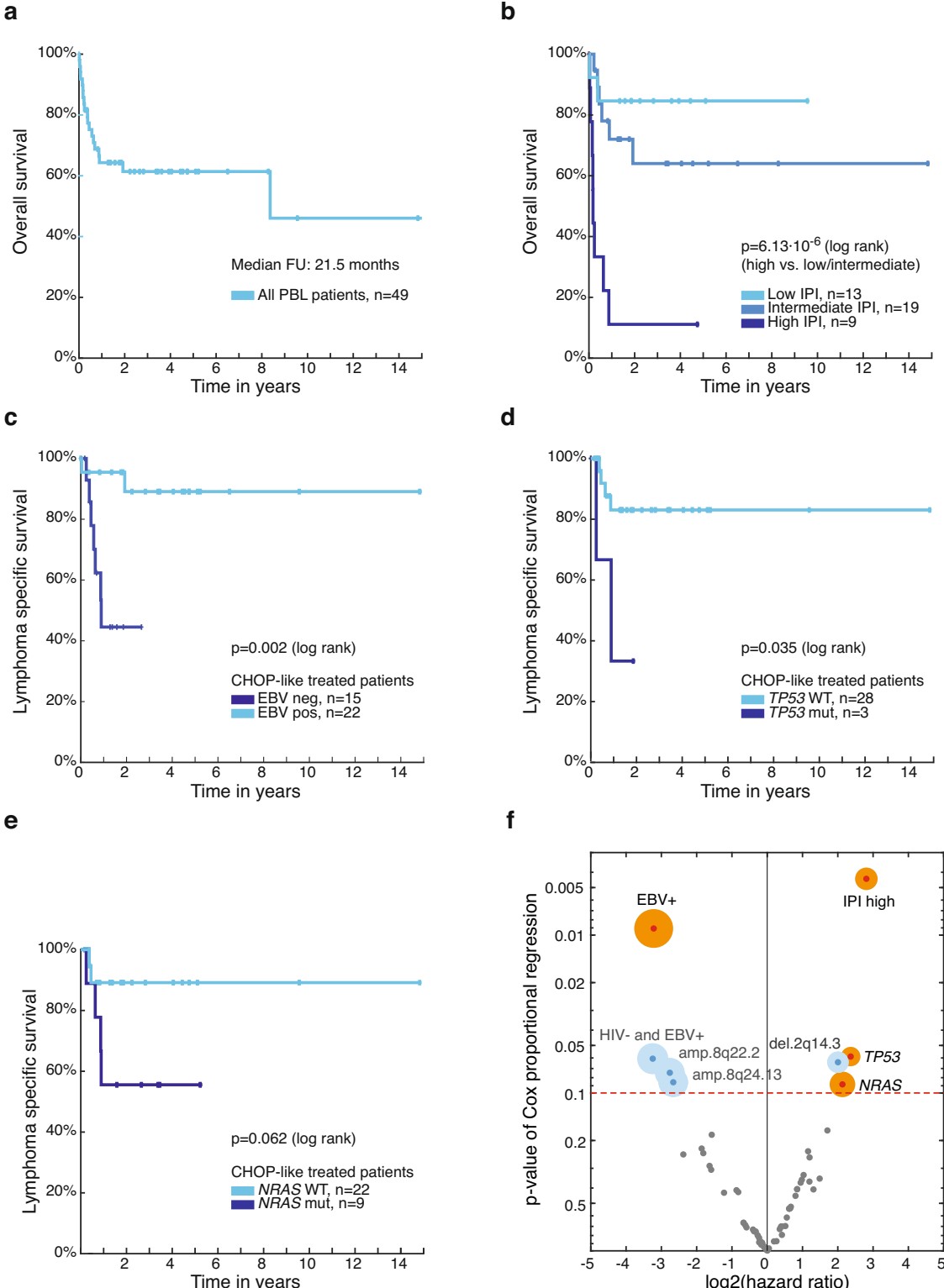

**Fig. 3 Survival analysis of PBL patients.** Kaplan–Meier (KM) estimates showing **a** overall survival (OS) for all PBL patients with available follow-up (FU) (*n* = 49) and **b** OS according to IPI group (IPI low: 0–1, IPI intermediate: 2–3, IPI high: 4–5). Lymphoma-specific survival (LSS) shown for PBL patients treated with CHOP-like chemotherapy depending on **c** EBV infection status of lymphoma (neg negative, pos positive), on **d** mutational status of *TP53*, and **e** *NRAS* (wt wildtype, mut mutated). Corresponding *p* values of two-sided log rank tests are shown, for multiple hypothesis testing over preselected conditions see Supplementary Data 9. **f** Hazard ratios by Cox regression and corresponding $p_{Cox}$ values describing the impact of significant mutations, SCNAs (with $q_{M2CV/G2.0}$ < 0.1) or clinical metadata on LSS for PBL patients treated with CHOP-like chemotherapy. The size of the circle for parameters with *p* value <0.1 is proportional to the number of affected patients. Earlier findings confirmed by this unbiased exploratory analysis are highlighted in orange. No significant hypotheses with FDR *q* < 0.1 were found.

diseases, organ transplantation, or other viral infections such as chronic hepatitis C infection (Supplementary Data 1). We detected focal deletions of 4q35.2 (50% vs. 11%, $p = 0.0035$, $q = 0.0671$) and broad deletions of 18p (33% vs. 4%, $p = 0.0083$, $q = 0.0788$) significantly more frequently in immunocompetent compared to immunocompromised patients. Recurrent amplifications or mutations did not significantly differ.

Possibly also due to limited case numbers, we did not reveal any significant differences comparing PBL patients with high IPI vs. patients with low/intermediate IPI as well as comparing patients with LSS of less than 12 months vs. patients with LSS of more than 24 months.

**Dissecting the IRF4 and STAT3 pathways in a plasmablastic cell model.** To functionally validate the identified oncogenic drivers and to potentially detect previously unappreciated dependencies that might guide targeted treatment approaches for PBL patients, we performed an unbiased shRNA screen using a customized shRNA library comprising 2669 different shRNAs targeting 768 genes in the only available PBL cell line PBL-1[21]. PBL-1 has been derived from an HIV positive PBL patient and represents an adequate functional model showing the typical plasmablastic immunophenotype and characteristic genetic lesions[21] (Supplementary Data 1, 4, and 8). Our shRNA screen was performed in duplicates. After 12 days of shRNA induction, shRNA abundance was determined by next generation sequencing. The screen was conducted in the manner that cells expressing shRNAs directed against oncogenes promoting cell proliferation and cell survival are depleted from the cell population. Our screen revealed that shRNAs directed against MYC (2 out of 2), IRF4 (4 out of 4), and STAT3 (1 out of 1) were among the most significantly depleted shRNAs (Fig. 4a and Supplementary Data 10). Since IRF4 and components of the JAK-STAT pathway were recurrently activated by genetic alterations in our PBL patient cohort, we focused on these two targets.

IRF4 protein expression was detectable in every sample of our cohort and focal IRF4 amplifications were identified in 29% of samples, both suggesting an involvement in the pathogenesis of PBL. First, we confirmed the results of the shRNA screen using two previously described and independent shRNAs[22,23]. As expected, both IRF4 shRNAs significantly downregulated IRF4 expression (Supplementary Fig. 10a). Transduction of these shRNAs induced cytotoxicity in PBL-1 cell and in the DLBCL cell lines OCI-Ly10 from the activated B-cell subtype (ABC), whereas germinal center B-cell-like (GCB) DLBCL models, used as negative controls, were unaffected (Fig. 4b). These results confirm an addiction to IRF4 signaling in PBL-1 cells. To utilize this addiction therapeutically, we treated PBL-1 cells with lenalidomide, previously been shown to downregulate IRF4[24]. Indeed, treatment with lenalidomide significantly downregulated IRF4 expression (Fig. 4c) and induced toxicity in PBL-1 cells. As previously shown, the viability of OCI-Ly10 was significantly inhibited by lenalidomide, but not in GCB DLBCL cells that were used as negative controls[24] (Fig. 4d). These results suggest that lenalidomide could be used therapeutically in PBL patients.

Next, we investigated the therapeutic potential of addiction to the JAK-STAT pathway. As described in 25% of primary PBL samples, PBL-1 cells also harbor a missense mutation of STAT3 within the SH2 domain (Supplementary Data 4) and PBL-1 represents an IL-6 dependent cell line[21]. First, we confirmed the results of the shRNA screen by two different shRNAs (Supplementary Fig. 10b). In line with the screening data, knockdown of STAT3 was selectively toxic to PBL-1 cells, whereas control DLBCL cell lines were unaffected (Fig. 4e). Consistently, the STAT3 antisense oligonucleotide AZD9150

induced significant toxicity specifically in PBL-1 cells, whereas DLBCL control cells remained unaffected (Supplementary Fig. 10c and Fig. 4f).

To analyze the functional consequences of the STAT3 mutations, we introduced either the p.D661Y STAT3 mutation representing the most frequently detected STAT3 mutation in our patient cohort or wild-type STAT3 in the GCB DLBCL cell line HT that lacks constitutive STAT3 signaling. Expression of wild-type STAT3 increased STAT3 signaling measured by phosphorylated STAT3 (pSTAT3) in Western blot to a significantly lesser degree compared to the p.D661Y STAT3 mutation (Fig. 4g). These results suggest that STAT3 mutations induce constitutive STAT3 signaling, possibly by sensitizing STAT3 for upstream activation. Next, we investigated whether PBL-1 cells require signaling through the p.Q643R mutation for survival. To this end, PBL-1 cells were retrovirally engineered to express either wild-type or mutated STAT3 (p.Q643R) (Supplementary Fig. 10d). In contrast to wild-type STAT3, p.Q643R STAT3 mutations rescued PBL-1 cells partially from STAT3 shRNA-induced toxicity, suggesting that PBL-1 cells are addicted to p.Q643R-induced oncogenic signaling (Fig. 4h). Finally, we investigated whether PBL-1 cells, despite harboring the activating STAT3 mutation, still depended on an upstream stimulus[21]. Removing interleukin-6 (IL-6) from the medium led to a significant decrease of pSTAT3 measured by Western blotting indicating that cells harboring STAT3 mutations still require an upstream signal (Fig. 4i). Treating PBL-1 with the pan-JAK inhibitor tofacitinib led correspondingly to a decrease of pSTAT3 levels (Fig. 4j) and significantly decreased cell viability suggesting a potential therapeutic use for PBL patients (Fig. 4k).

## Discussion

Our study represents a comprehensive genetic analysis of a large cohort of primary PBL samples of all subtypes. Several identified genetic alterations represent directly targetable vulnerabilities that might guide novel therapeutic strategies for PBL patients. Roughly one half of PBL cases were characterized by recurrent mutations of genes encoding components of the oncogenic RAS-RAF signaling pathway. Detected NRAS mutations occurred exclusively at the known hotspot residues p.G12, p.G13, and p.Q61 representing gain of function mutations in various cancer entities including multiple myeloma (MM)[25–27]. Interestingly, RAS mutations are rare in other aggressive lymphoma subtypes such as DLBCL or Burkitt lymphoma[28–31]. While direct targeting of RAS proteins remains challenging, functional studies should address the therapeutic potential of BRAF/MEK/ERK inhibition or involved downstream pathways[32]. In MM, targeting the BRAF V600E mutation seems to be promising[33] and clinical trials currently evaluate the combination of BRAF and MEK inhibitors in affected patients.

While STAT3 mutations have been described in 40% of T-cell large granular lymphocytic leukemia (T-LGL)[34], in 30% of chronic lymphoproliferative disorders of natural killer cells[35], and in roughly 10% of subsets of T-cell lymphoma[36], STAT3 mutations are rare in B-cell lymphomas[28,37,38]. Intriguingly, 25% of analyzed PBL cases harbored STAT3 mutations that clustered mainly in the SH2 domain that is essential for dimerization and activation of STAT3[10]. The identical mutational hotspots p.D661 and p.Y640 were previously detected in primary samples of T-LGL[34]. In contrast, mutations of the JAK-STAT pathway have not been recurrently detected in MM[38–40]. Our functional analyses using an unbiased shRNA screen performed in the PBL cell line PBL-1 showed that STAT3 indeed represents an attractive molecular target. Our functional data suggest that STAT3 mutations induce constitutive STAT3 signaling, possibly by sensitizing

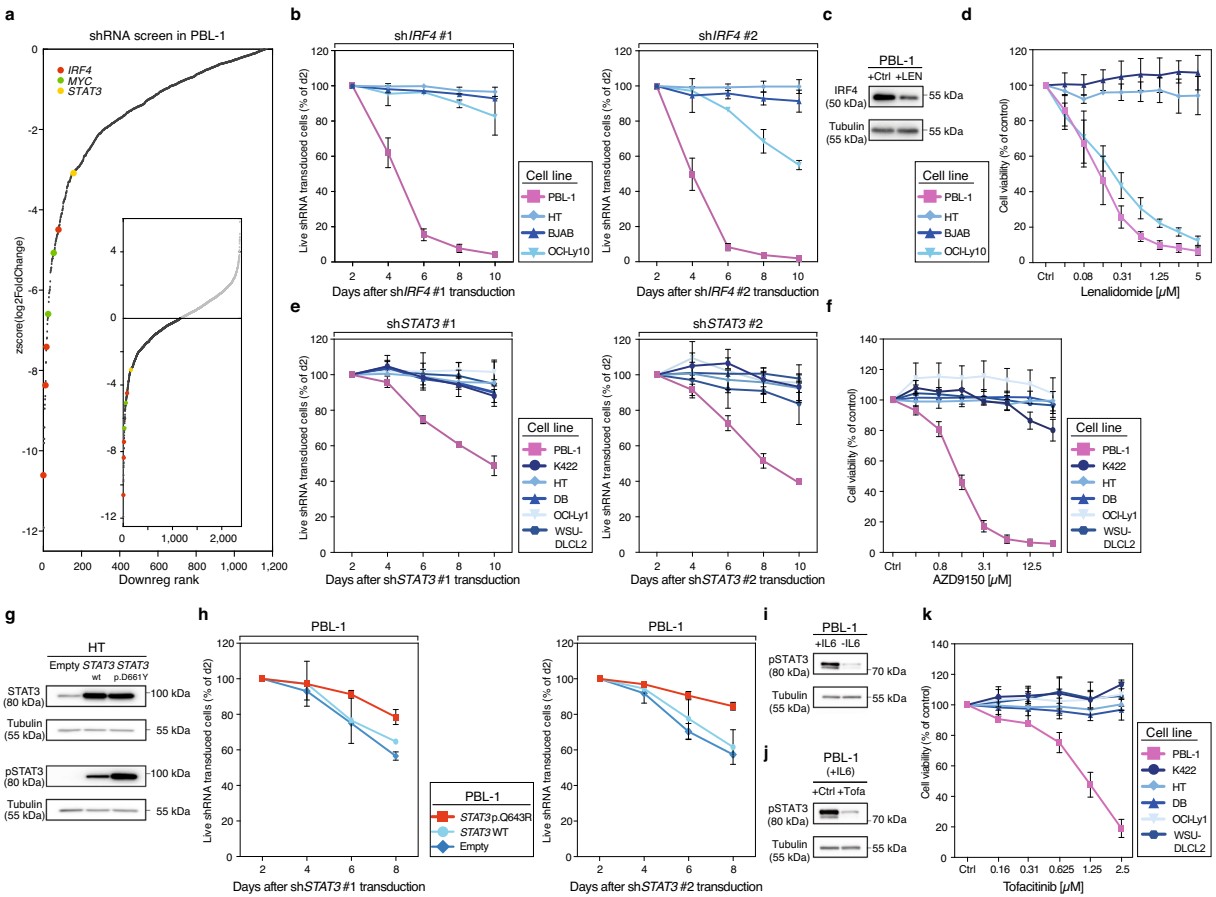

**Fig. 4 Identification of molecular targets for the treatment of PBL. a** shRNAs used in the shRNA library screen performed in PBL-1 cells are sorted and ranked by the corresponding *z*-score of log2-transformed depletion ratios after 12 days (see Supplementary Data 10 for raw read counts). shRNAs targeting *IRF4*, *MYC*, and *STAT3* are highlighted. **b** The graphs show the percentage of viable shRNA *IRF4* #1/#2-transduced cells over time normalized to control shRNA and to day two fraction. Mean values ± standard deviations are shown of three experiments. **c** IRF4 protein expression by Western blot (WB) in PBL-1 cells treated with DMSO (Ctrl) or lenalidomide (LEN; 1.25 μM) for 24 h respectively. Representative results are shown of three independent experiments. **d** Cell viability after treatment with indicated concentrations of lenalidomide for 5 days normalized to DMSO-treated cells. Mean values ± standard deviations are shown of three experiments. **e** Effect of *STAT3* knockdown by two independent shRNAs on cell viability of PBL-1 cells over time. Data were shown as means ± standard deviations of three experiments. **f** Relative cell viability after treatment with increasing concentrations of the *STAT3* antisense oligonucleotide AZD9150 for 5 days. Data were shown as means ± standard deviations of three experiments. **g** Phosphorylated STAT3 levels determined by WB following transduction with a control plasmid (empty), wild-type (wt) *STAT3*, and p.D661Y *STAT3* cDNA in HT cells. Representative results are shown of three independent experiments. **h** Exogenous expression of the p.Q643R *STAT3* cDNA rescues PBL-1 cells from *STAT3* shRNA-mediated toxicity. In contrast, a control plasmid (empty) or wild-type (wt) *STAT3* did not induce a rescue of *STAT3* shRNA-transduced PBL-1 cells. Data were shown as means ± standard deviations of two experiments. **i** Phosphorylated STAT3 levels determined by WB in PBL-1 cells supplemented with or without IL-6 (5 ng/ml) or **j** treated with the pan-JAK inhibitor tofacitinib (0.5 μM) for 4 h. Representative results are shown of three independent experiments. **k** The pan-JAK inhibitor tofacitinib decreases the cell viability of PBL-1 cells. In contrast, DLBCL control cell lines are unaffected by tofacitinib treatment. Data were shown as means ± standard deviations of three independent experiments. Source data for Fig. 4b–k are provided as a Source Data file.

STAT3 for upstream activation. Comparable functional studies have been reported in T-LGL[34]. This might be a rationale for the use of JAK inhibitors or IL-6-antagonists in the treatment of PBL patients.

Our analyses in PBL subcohorts further showed that *STAT3* mutations predominantly occur in HIV-associated PBLs as these lymphomas harbored significantly more frequently *STAT3* mutations compared to other PBL subtypes. In line with our data, a recently published analysis of HIV-associated PBLs reported *STAT3* mutations at the same identified mutational hotspots in 42% of investigated cases[9].

Another genetic hallmark of PBL represents *MYC* translocations that are detectable in roughly 50% of cases, corresponding well to the previous reports[7]. *MYC* mutations occurred only in translocated cases, probably due to somatic hypermutation[14], while functional consequences still need to be elucidated.

Additionally, *MYC* might be activated through different molecular mechanisms in PBL as the recurrently altered RAS-RAF, JAK-STAT, IRF4, and NOTCH signaling pathways are known to activate *MYC* as downstream target[22,23,41–44].

Our Oncoscan analyses identified recurrent SCNAs in PBL. Specifically, we identified frequent broad alterations of chromosomes 1q and 7 as well as several specific focal aberrations. *IRF4* was affected by focal amplifications of 6p25.3. Our shRNA screen showed that shRNAs targeting *IRF4* appeared among the most toxic ones in our plasmablastic cell line model. Correspondingly, treatment with lenalidomide, that has been shown to down-regulate IRF4 expression[24], induced cytotoxicity in PBL-1 cells confirming promising case reports using lenalidomide for the treatment of patients with chemorefractory PBL[45,46].

PBL can represent a diagnostic challenge. This includes the differentiation from some MM cases. Osteolytic bone lesions and/

or high paraprotein favor the diagnosis of MM while latent EBV infection of lymphoma cells and HIV infection of affected patients remain rare[47,48]. Gains and amplifications of 1q and chromosome 7, as well as deletions of chromosomes 13q, are shared by both entities[49,50]. Multiple trisomies typical of the hyperdiploid MM subtype were not detectable in PBL[49]. The highly characteristic amplification of *IRF4* (6p25.3) has not yet been reported in MM at a higher frequency, although myeloma cells are functionally addicted to IRF4 signaling and translocations involving 6p25.3 have been described[23,51]. *RAS* mutations have been recurrently detected in both entities, but frequent mutations in MM such as *FAM46C* or *DIS3* were uncommon in our PBL cohort[39]. At last, recurrent mutations of genes encoding components of the JAK-STAT pathway have not been detected in MM. These different aberrations might help in differentiating PBL from MM in the future if tumor samples are analyzed for these alterations in the clinical setting.

Even more difficult can be the distinction of PBL from extramedullary plasmablastic myeloma as they can be histologically indistinguishable[52]. For these rare cases, clinical parameters and often the clinical course currently guide diagnosis and subsequent therapy. Apart from very small studies, the genetic landscape of plasmablastic myeloma is largely unknown. However, a small targeted sequencing analysis identified alterations of *MYC* and of genes encoding components of the RAS-RAF pathways, while aberrations affecting the JAK-STAT signaling were not detectable[53]. Nevertheless, significantly larger and more comprehensive analyses are warranted to identify the molecular differences and similarities between PBL and plasmablastic myeloma.

For 49 patients we were able to obtain survival data enabling us to investigate the impact of specific genetic aberrations on the survival of affected patients. Our survival analyses revealed that the IPI provides a valid risk stratification for PBL patients. High-risk patients showed poor outcome following chemotherapy suggesting that this patient subgroup is in special need of new therapeutic approaches. Moreover, patients with EBV negative disease were characterized by significantly inferior survival supporting a retrospective analysis of 135 PBL cases of the LYSA group[3]. This finding might be related to the fact that EBV negative PBL cases seem to harbor more frequent mutations of *TP53* that might be associated with the chemorefractory disease. Correspondingly, PBL patients with lymphomas harboring *TP53* mutation showed an unfavorable clinical course.

In summary, we have identified previously unknown genetic alterations affecting the RAS-RAF, JAK-STAT, MCL1, IRF4, and NOTCH pathways. These insights will help to define PBL on a molecular level. Functional analyses of these aberrations identified particularly the IRF4 and the JAK-STAT pathways as therapeutically targetable vulnerabilities for the rational treatment of PBL patients. These promising preclinical findings warrant further clinical testing to improve the outcome of patients diagnosed with PBL.

## Methods

**Patient samples**. FFPE material of primary PBL samples was collected from the University Hospitals in Münster, Kiel, Würzburg, Berlin, Basel, Glasgow, Tübingen, the Robert-Bosch Hospital in Stuttgart, and the Germans Trias i Pujol Hospital, University Hospital of Bellvitge, Hospital Vall d'Hebron and Hospital del Mar in Barcelona, and Hospital Gregorio Marañón in Madrid. Selected primary PBL cases were independently reevaluated in a central pathology review of four expert hematopathologists (Ioannis Anagnostopoulos, Wolfgang Hartmann, German Ott, and Gustavo Tapia). Individual cases that were subsequently included in our analysis were evaluated by at least two different expert hematopathologists. From initially 118 collected cases, 96 cases were histologically confirmed as PBL according to the criteria of the WHO Classification of 2017[2] (Supplementary Fig. 1a and Supplementary Data 1). Twelve cases were excluded as plasma cell neoplasm, immunohistochemical staining revealed four cases to be ALK + DLBCL

and three cases to be HHV8 + lymphoproliferation. The tumor content was microscopically determined on hematoxylin and eosin-stained slides. Specifically, the relative percentage of tumor blastic B-cells was recorded in increments of 10% taking into account the relative percentages of non-blastic bystander lymphocytes and histiocytes (Supplementary Data 1). The median tumor content was 90% (min–max range: 20–100%). For 22 PBL specimens, paired normal tissue was available. Sites of origin are listed in Supplementary Data 1. Contamination with tumor cells was excluded based on conventional hematoxylin/eosin and immunohistochemistry. For 49 cases with molecular data corresponding clinical and survival data were retrieved (Supplementary Data 1). This study was approved by institutional ethics review boards of the University Hospitals in Tübingen, Glasgow, and Basel and the Germans Trias i Pujol Hospital in Barcelona, in accordance with the Declaration of Helsinki. Informed consent was obtained according to the requirements of the responsible ethics committee.

**DNA extraction**. DNA was extracted from FFPE samples using the Gene Read DNA FFPE Kit (Qiagen, Hilden, Germany). To assess quality, we applied a qPCR-based method by Illumina (Illumina FFPE QC kit, San Diego, USA) comparing the amplificability of extracted DNA to a reference template. According to Illumina, obtained ΔCt values (cycle threshold) indicate good DNA quality when a value of ≤2 was determined. DNA concentration was determined using the Qubit DNA Quantification Assay Kit (Thermo Fisher Scientific, Waltham, Massachusetts, USA).

**Immunohistochemistry and FISH**. To achieve homogenous immunophenotype data, tissue microarrays were constructed from 68 suitable PBL FFPE cases[54]. In detail, three 0.6 mm thick cores per case were taken from the selected donor block using a "manual tissue puncher" (Beecher Instruments, Silver Spring, Maryland, USA) and inserted into the recipient block. Immunohistochemical stainings for CD3, CD19, CD20, CD30, CD38, CD56, CD138, PAX5, MUM1/IRF4, BLIMP1, Ki-67, MYC, HHV8, ALK, PD-L1, PD-L2, and EBER in situ hybridization were performed according to established protocols[54] (Supplementary Table 2). The percentage of positive tumor cell nuclei was recorded for BLIMP1 and Ki-67 in increments of 10%. All other markers were scored to be positive or negative regardless of staining intensity according to a predefined cutoff (Supplementary Data 1).

FISH was performed according to standard procedures[55,56]. The 57 PBL specimens were hybridized with the Vysis LSI *MYC* dual color break-apart probe (BAP) (Abbot Molecular, Wiesbaden, Germany) and 63 cases with the *MYC-IgH* fusion probe (FP) (Abbot Molecular) (Supplementary Data 1). The evaluation was performed according to standard procedures[57]. At least 100 intact nuclei per case were evaluated using an epifluorescence microscope (Leica Microsystems, Bensheim, Germany). Images were captured using the ISIS imaging system (MetaSystems, Altlussheim, Germany) (Supplementary Fig. 2).

### WES and data analysis

*Library generation*. For each sample, we provided DNA amounts as recommended by the FFPE QC Kit (Illumina), a qPCR-based method comparing the ability of amplification for extracted DNA to a reference template (Supplementary Data 1). We correlated age of FFPE material with obtained ΔCt values and found a significant correlation of higher age with lesser DNA quality expressed by higher ΔCt values ($r = 0.53$, $p = 1.5 \times 10^{-7}$) (Supplementary Fig. 11a). For exome capturing, the Agilent SureSelect Human All Exon V6 Kit (Agilent Technologies, Santa Clara, California, USA) was applied according to the manufacturer's instructions. In brief, DNA samples were sheared, end-repaired, ligated with adapter molecules, and amplified. Magnetic beads were used to capture exonic DNA regions. Captured libraries were enriched, purified, and quantified.

*Sequencing and alignment*. Sequencing was performed on a HiSeq platform (Illumina) with 150 bp paired-end reads. Measured sequence reads were aligned against the current human reference genome from the Genome Reference Consortium (GRCh38) using HISAT2 v2.1.0[58,59]. We verified sample concordance using the NGSCheckMate program and confirmed that each tumor/normal pair had a matching germline[60].

*Quality control on read and sample level, effective coverage, and variant discovery*. For variant discovery, we utilized the Genome Analysis Toolkit (GATK) v4.1.2.0[61] and Mutect 2.1[62]. To determine the effective coverage, we counted only reads aligned by HISAT2 that also passed GATK and Mutect read level quality control read filters, i.e., only those reads that were actually utilized for subsequent variant discovery (see Supplementary Fig. 4 for the distribution of effective coverage). Three samples were excluded due to low effective coverage and one further sample due to hypermutation (4/89).

*Basic variant filtering*. To build a panel of normal variants (PON), we first performed variant discovery with the same experimental and analytical pipeline for 38 normal controls (comprised of 29 normals from patients and additional nine samples from healthy donor lymph nodes). A variant was included in the PON if Mutect determined it as significant in at least two independent subjects. This PON

was subsequently used to filter germline variants and potential pipeline-specific artifacts by applying Mutect. Additionally, we used the gnomAD database as a large population germline resource based on the Exome Aggregation Consortium ExAC[63]. For tumor samples for which DNA-sequencing of paired normal cell samples were available, we additionally utilized these specimens for a more specific paired statistical variant analysis by Mutect. Otherwise, we used the unpaired analysis mode.

*Advanced variant annotation and filtering.* Next, we applied an optimized multi-stage filter hierarchy to reach the maximal specificity of somatic mutation calls. All filter steps in the applied order are listed in Supplementary Data 2. For this hierarchy, we first annotated discovered variants with their transcript and protein level consequences using TransVar 2.4.1[64] and the NCBI RefSeq gene models[65]. In the case of multiple RefSeq transcripts per gene, we annotated each variant with the one leading to the strongest possible biological consequence on protein level according to TransVar. For mutation overview plots, we selected the first principal transcript of the respective gene according to the APPRIS database[66]. Additionally, we annotated variants with confirmed somatic mutations according to the Catalog Of Somatic Mutations In Cancer (COSMIC v85[67]), the NCBI database of common human variants (≥5% in any of the five large populations from dbSNP build 151[68]), and NCBI ClinVar[69] (version 2018-04) using vcfanno v0.3.0[70]. To filter FFPE specific artifacts, we fitted the read orientation model of GATK 4.1.2 (LearnReadOrientationModel) that flags variants with a significant bias between forward and reverse reads. As first countermeasure for alignment artifacts caused by high sequence homology, e.g., from pseudogenes, we additionally filtered variants utilizing the GATK FilterAlignmentArtifacts model. We correlated the number of called somatic mutations to the quality of DNA (measured by $\Delta$Ct values) without finding any significant correlation ($r = -0.02$, $p = 0.84$; Supplementary Fig. 11b) indicating that our filter hierarchy recognizes specific FFPE artifacts efficiently and compensates for lower quality of provided DNA. Microscopically determined tumor load did also not correlate with the number of called somatic mutations ($r = 0.04$, $p = 0.75$; Supplementary Fig. 11c).

*Variants called somatic and specificity gains by paired samples.* Based on variant statistics from Mutect, GATK, and all annotations, our filter hierarchy called 0.09% of all variants as somatic mutations (see Supplementary Data 2 for detailed mutation counts and percentages remaining after each filtering step). To compare pipeline specificity loss between samples having patient-matched normals and tumor-only samples, we recomputed paired initial tumor samples in unpaired mode (i.e., without using their matched germline normals). We determined a sensitivity of 0.98, a precision of 0.56, and a balanced *F*-score of 0.72.

*Gene level mutation analysis.* Based on called somatic mutations (and, for background estimation, somatic variants that are synonymous with respect to protein level consequences), potential cancer driver genes were predicted using MutSig2CV version 3.11[71]. In addition to variant level filtering and since high sequence homology causes certain genomic regions to be impossible to analyze with current WES technology[72], we filtered for genes that still had variants with artificially high coverage (such as several *MUC\** genes) (Supplementary Fig. 4). Next, we created MAF-formatted input files in HG19 coordinates for MutSig2CV using CrossMap version 0.4.0[73]. Finally, we considered all top genes with a MutSig2CV false discovery rate of $q < 0.1$ for subsequent analyses (Supplementary Data 3).

*Additional tools and software utilized for WES analysis.* For various analysis tasks in the sequencing pipeline, we used bedtools[74], the Integrated Genomics Viewer (IGV v2.6.3-2.8.0)[75], the Picard toolkit (https://broadinstitute.github.io/picard/), and SAMtools[76]. For analysis pipeline orchestration including parallel remote analysis jobs on high-performance clusters as well as for most visualizations including oncoplots, we used MATLAB® (versions R2018a-R2020a, The MathWorks® Inc., Natick, Massachusetts, USA). We used Microsoft Excel (versions 2016-2019) for collecting clinical metadata and presenting results. R (version 3.6.3, R Foundation for Statistical Computing, Vienna, Austria), Python (version 2.7-3.X, Python Software Foundation, Wilmington, Delaware, USA), and GNU parallel[77] were used for running various tools or for local parallelization. TMB was computed and visualized together with TCGA data using maftool v2.7.41.[78] Needle plots of mutation profiles were created using ProteinPaint[79]. Based on protein structures from the PDBe-KB database[80] that correspond to principal transcripts selected by APPRIS, we visualized the location of mutations and their spatial clustering in 3D protein structures with PyMOL[81]. All used tools are summarized in Supplementary Table 3.

*Targeted resequencing.* Targeted resequencing was performed by Next Generation Sequencing (Ion GeneStudio S5 prime, Thermo Fisher Scientific) using an AmpliSeq Custom Panel comprising hotspot regions in *STAT3*, *KRAS*, *NRAS*, *BRAF*, and the complete coding sequence of *TP53* and *TET2* (Supplementary Data 6). Amplicon library preparation and semiconductor sequencing were done according to the manufacturers' instructions using the Ion AmpliSeq Kit for Chef DL8, the Ion 510 & Ion 520 & Ion 530 Kit—Chef, the Ion 530 Chip Kit on the Ion Chef, and the Ion GeneStudio S5 Prime system (Thermo Fisher Scientific). Output files were generated with Torrent Suite 5.12.0. Variant calling of non-synonymous

somatic variants compared to the human reference sequence was performed using the Ion Reporter Software (Thermo Fisher Scientific, Version 5.16.0.2). Variants called by the Ion Reporter Software were visualized using the Integrative Genomics Viewer (IGV 2.8.0)[75] to exclude panel-specific artifacts.

## Determination of SCNAs

*Sample preparation and SNP microarray measurements.* To determine SCNAs, 80 ng of extracted DNA was used applying the Oncoscan CNV FFPE Assay Kit (Affymetrix, Thermo Fisher Scientific) according to the manufacturer's instructions. In brief, the Oncoscan assay uses a microarray technique consisting of over 220.000 molecular inversion probes (MIP). MIP bind to target DNA forming an incomplete, circular loop and leaving a gap at a specific SNP site. Following annealing, probes are distributed to wells either containing adenosine and thymidine triphosphate or guanosine and cytosine triphosphate nucleotides. Uncircularized MIP and genomic DNA are digested by exonucleases so that only closed, circular MIP remains. MIP are linearized and amplified. Finally resulting fragments bind to the Oncoscan assay array, are fluorescently stained and visualized in the GeneChip™ Scanner 3000 7 G (Thermo Fisher Scientific). Fluorescence is proportional to the copy number of analyzed genomic sites.

*Sample level copy number segmentation and purity.* Oncoscan raw data files were preprocessed using the Chromosome Analysis Suite (ChAS version 4.0, Thermo Fisher Scientific). Measurements by individual SNP probes were aggregated to segments of unchanged allele-specific copy numbers by ASCAT v2.4.3[16]. ASCAT also estimated sample ploidy and purity (i.e., the cell fraction originating from aberrated tumor cells as opposed to non-aberrated bystander cells).

*Cohort level SCNA discovery.* Recurrent SCNAs were identified and statistically evaluated by GISTIC v2.0.23[15]. Visual quality control of sample counts having a particular SCNA over the genome axis revealed that some peak regions were too narrow (for example in the *IRF4* locus, the region focused on a ~0.3 Mb peak of three samples on top of a well-defined ~0.4 Mb peak of ten additional samples above baseline) (Fig. 2b). As this could not be solved by adjusting GISTIC's RegBounder confidence parameter, we applied a robust peak extension around GISTIC-determined MCRs. First, we estimated from broad SCNAs the local baseline counts $n_{base}$ of samples that have the current aberration type at peak site (by averaging sample counts over all SNP probes in the MCR ± 20% of the chromosome arm length). Next, we counted samples at each SNP probe in the GISTIC MCR and took their maximum $n_{max}$. With $n_{peak} = n_{max} - n_{base}$ counting samples that have this aberration above baseline, we extended the MCR to the left and right as long as $\geq 40\% \cdot n_{peak}$ samples having this aberration type remained. This cuts well-defined peaks at their flanks. To avoid broad uphill extensions, we additionally stopped extension once counts grew $\geq 80\% \cdot n_{peak}$ after they had already fallen below this threshold between the MCR and the current extension position.

**Mutation clonality by integrative analysis of WES and SNP results**. To identify potential early mutations in the pathogenesis, we estimated mutation clonality. First, we integrated Oncoscan results (tumor purity, ploidy, and copy numbers) with variant AFs from the WES data to estimate cancer cell fractions (CCF) that harbor specific mutations. We utilized the formula[82]:

$$f_{CCF} := \frac{f_{VAF}}{f_{purity}} \cdot \left( \left( 1 - f_{purity} \right) \cdot n_{CN,normal} + f_{purity} \cdot n_{CN,tumor} \right) \qquad (1)$$

where $f_{VAF}$ denotes variant AF, $f_{purity}$ denotes tumor purity, $n_{CN,normal} = 2$ assuming diploid normal bystander cells, and $n_{CN,tumor}$ denotes tumor cell copy numbers as estimated by ASCAT. Then, we defined clonal variants using the established threshold of $f_{CCF} \geq 0.9$[28].

**Statistical association analyses**. We tested for associations between gene mutations and SCNAs in the following defined clinical subcohorts: EBV positive vs. EBV negative disease, HIV positive vs. HIV negative patients, HIV positive patients with EBV positive PBL vs. HIV positive patients with EBV negative PBL, HIV negative patients with EBV positive PBL vs. HIV negative patients with EBV negative PBL, PBLs arising in the oral cavity/nasopharynx vs. PBLs arising elsewhere, *MYC* translocated vs. untranslocated, patients with vs. without immunosuppression, patients with high IPI vs. low/intermediate IPI, patients with lymphoma-specific survival (LSS) of less than 12 months vs. patients with LSS of more than 24 months, CD20 negative PBLs vs. weakly CD20 positive cases. To test for a significantly higher median mutation/SCNA count in one subcohort vs. the other, we utilized a one-tailed Wilcoxon rank-sum test for each selected genetic lesion (Supplementary Data 5). We focused our hypotheses on the identical gene set of interest as shown in Fig. 1a (cohort mutation frequency ≥5%, $q_{M2CV} < 0.1$ plus *MYC* and *BRAF*) and the recurrent SCNAs reaching significance according to GISTIC as shown in Fig. 2 (cohort frequency ≥5%, $q_{G2.0} < 0.1$). FDR were computed using the Benjamini and Hochberg (BH) method[83]. For significance, we used a prescribed error threshold of $q < 0.1$.

Likewise and for control purposes, we compared mutation frequencies between the subcohorts of PBL cases with matched normal vs. PBL cases without matched normal for all genes having a cohort frequency ≥10% (Supplementary Data 5).

**Survival analyses**. We analyzed OS and LSS as clinical endpoints. OS was defined as the time from diagnosis until death from any cause. Subjects not recorded to be dead were censored at the follow-up time last known to be alive. LSS was defined as the time from diagnosis until death due to lymphoma. Patients alive or dead from other reasons were censored at their last follow-up time.

We estimated OS for the full cohort and compared OS between different risk groups according to IPI. To assess survival differences between any two patient subgroups, we computed log rank tests. We chose LSS as an endpoint to test the following biologically preselected conditions: IPI, the status of EBV and HIV, translocation and expression of *MYC*, amplification of 6p25.3 (*IRF4*), deletion of 17p, significant mutations ($q_{M2CV} < 0.1$) with cohort frequency >10% (*NRAS*, *KRAS*, *STAT3*, and *TP53*). FDRs were again computed using the BH method with a significance threshold of $q < 0.1$ (Supplementary Data 9)[83]. As an additional independent exploratory analysis, we tested for association between LSS and all significant mutations ($q_{M2CV} < 0.1$), all significant SCNAs ($q_{G2.0} < 0.1$) as well as clinical metadata, using univariate Cox proportional hazard regression, FDRs were calculated according to BH (Supplementary Data 9)[83].

**Cell culture, retroviral constructs, and cytotoxicity assay**. The plasmablastic cell line PBL-1 was cultured in RPMI 1640 with 20% fetal calf serum (FCS) and supplemented with interleukin-6 (IL-6; 5 ng/ml)[21]. DLBCL cell lines HT, Karpas 422 (K422), DB, WSU-DLCL2, and BJAB were grown in RPMI 1640 with 10% FCS, OCI-Ly1, and OCI-Ly10 in Iscove's Modified Dulbecco's medium supplemented with either 10% FCS or 20% human plasma[84]. All cells were maintained at 37 °C and 5% $CO_2$.

All cells were modified to express a murine ecotropic receptor and the rtT3 tetracycline transactivator[85–87]. Transduction of small hairpin RNAs (shRNAs) or complementary DNA constructs (cDNA) was performed[18,85]. The targeting sequences of *IRF4* shRNAs #1 and #2 were CCGCCATTCCTCTATTCAAGA and GTGCCATTTCTCAGGGAAGTA[22]. The sequences of *STAT3* targeting shRNAs #1 and #2 were GCCACTTTGGTGTTTCATAAT (binding in the coding sequence) and GCATAGCCTTTCTGTATTTAA (binding in the 3′-untranslated region), respectively (Supplementary Table 4). To assess shRNA-mediated cytotoxicity assay, retroviruses that coexpress GFP were used. In brief, flow cytometry (AttuneNxT, Thermo Fisher Scientific) was performed 2 days after retroviral transduction to determine the initial GFP-positive proportion of live cells. Cells were subsequently cultured in a medium with doxycycline to induce shRNA expression and sampled over time. The GFP-positive proportion at each time point was normalized to a nontoxic control shRNA[18] and additionally normalized to the initial GFP value. To rescue from toxicity after specific *STAT3* knockdown, we transfected PBL-1 cells with a *STAT3* cDNA construct. To prevent knockdown of the exogenous *STAT3* cDNA by shRNA#1, we introduced several silent mutations within the shRNA binding site in the *STAT3* cDNA applying the II Site-Directed Mutagenesis Kit (Agilent). Equally, we applied the II Site-Directed Mutagenesis Kit to introduce the p.D661Y mutant in *STAT3* cDNA for further functional analysis.

**shRNA library screen**. A customized shRNA library targeting 768 genes was designed and constructed[88,89]. In brief, a pool of 2669 shRNAs (Supplementary Data 10) was subcloned into a retroviral vector system allowing doxycycline-induced expression and was transduced into PBL-1 cells[89]. Following selection with blasticidin (150 µg/ml, InvivoGen, CA) for 5 days, day 0 (d0) samples were harvested and cells were subsequently cultured with and without doxycycline for 12 days. After 12 days, shRNA expressing cells (dsRed+/Venus+) were sorted by FACS (FACSAria III, BD, New Jersey, USA) and harvested (d12 samples). Genomic DNA of d12 was extracted using phenol and precipitated by isopropanol. Finally, sequencing libraries were created based on PCR amplification of shRNA guide strands and sequenced on a NextSeq platform (Illumina). Two biological replicates were analyzed. Reads were aligned using the BLAT algorithm (Standalone BLAT v.36)[90], and counts were aggregated for each shRNA sequence. After batch normalization and pre-filtering of shRNA reads with low coverage (sum of read counts at day12 on/off dox <50 [301 shRNAs]), we assessed the fold change and significance of shRNA depletion comparing the shRNA representation in samples after 12 days with vs. without induced shRNA using DESeq2[91] (for raw read counts, quality control details, and results, see Supplementary Data 10).

**Western blot**. Western blotting was performed according to standard procedures[92]. Cells were harvested and resulting protein lysates were quantified using the bicinchoninic acid assay (BCA) (Thermo Scientific). Proteins were electrophoresed on 10% SDS-PAGE gel and transferred to polyvinylidene difluoride membranes (Merck Millipore, Burlington, USA). Membranes were washed and stained with peroxidase-conjugated antibodies that specifically bind to the primary antibody. Binding was detected using Lumi-Light Western Blotting Substrate (Sigma Aldrich) and visualized using an Amersham Imager 600 (GE Healthcare Life Sciences, Chicago, USA). Primary antibodies directed against (p)

STAT3 (dilution 1:1,000), IRF4 (1:1,000), and Tubulin (1:5,000) were obtained from Cell Signaling Technology (Cambridge, UK) and Sigma Aldrich (St. Louis, Missouri, USA), respectively. PBL-1 cells were cultured in medium with and without IL-6 and were treated with 0.5 µM tofacitinib (Selleckchem, Houston, USA) or the corresponding amount of DMSO for 4 h at 37 °C (5% $CO_2$) until protein lysates were harvested and subjected to immunoblotting for (p)STAT3. PBL-1 cells were treated with 1.25 µM lenalidomide (Selleckchem) or the corresponding amount of DMSO for 24 h at 37 °C (5% $CO_2$) until protein lysates were harvested and subjected to immunoblotting for IRF4. To determine STAT3 knockdown following shRNA induction or treatment with AZD9150[93] (AstraZeneca, Cambridge, UK), cells were induced with doxycycline for 4 days or treated with 25 µM of AZD9150 for 1 day, respectively until protein lysates were harvested. To determine IRF4 knockdown following shRNA induction, cells were equally induced with doxycycline for 4 days until protein lysates were harvested.

**Cell viability assay**. PBL-1 and the DLBCL cell lines HT, K422, DB, WSU-DLCL2, BJAB, OCI-Ly1, and OCI-Ly10 were seeded in 96-well plates at 5000 cells per well. They were treated with different concentrations of the JAK inhibitor tofacitinib, the antisense oligonucleotide AZD9150, or lenalidomide (Selleckchem). Cell viability was measured on day 3 or 5 using the CellTiter-Glo Luminescent Assay (Promega, Dübendorf, Switzerland)[94]. Luciferase reaction is utilized to quantify the amount of ATP in viable cells. Produced luminescence was measured using the Victor X3 Plate Reader (Perkin Elmer, Waltham, Massachusetts, USA) and then compared with DMSO-treated cells or cells treated with an antisense oligonucleotide control molecule respectively.

**Reporting Summary**. Further information on research design is available in the Nature Research Reporting Summary linked to this article.

## Data availability

The whole-exome sequencing data EGAD00001006400 and SNP microarray data EGAD00010001978 generated in this study have been deposited in the European Genome-phenome Archive (EGA) under study accession EGAS00001004659. These data are available under restricted access for German data privacy laws; access can be obtained via the associated data access committee EGAC00001001735. The processed somatic mutations and copy number aberrations as well as clinical metadata and figure raw data are provided in respective Supplementary Data items or the Source Data file. The following public data sources were used in this study: The human reference genome from the Genome Reference Consortium (GRCh38) in its pre-indexed form for alignment with HISAT2[58,59] [http://daehwankimlab.github.io/hisat2/download/#h-sapiens], the Catalog Of Somatic Mutations In Cancer (COSMIC[67], v85) [https://cancer.sanger.ac.uk/cosmic], the NCBI database of common human variants (based on dbSNP build 151[68], version 2018-04) [https://www.ncbi.nlm.nih.gov/variation/docs/human_variation_vcf]), NCBI ClinVar[69] (version 2018-04) [https://www.ncbi.nlm.nih.gov/clinvar/], gnomAD/ExAC[63] germline variants as provided in the file af-only-gnomad.hg38.ensemble.vcf.gz of the GATK[61] resource bundle [originally accessed via ftp.broadinstitute.org/bundle, but since moved by the Broad Institute to Google cloud bucket; see https://gatk.broadinstitute.org/hc/en-us/articles/360035890811-Resource-bundle for access information], the PDBe-KB[80] for 3D protein information [https://www.ebi.ac.uk/pdbe/pdbe-kb], and the principal splice isoforms database[66] (APPRIS, version 2020-01-22) [https://github.com/appris/appris]. Source data are provided with this paper.

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

## Acknowledgements

We thank all patients and their physicians for trial participation. We thank Jan Mitschke who contributed to the bioinformatical analysis of the performed shRNA screen. This work was funded by a grant from German Cancer Aid to J.N.H. We thank the Instituto de Salud Carlos III, Ministerio de Economia y Competitividad in Spain as well as the Josep Carreras International Foundation for funding to J.-T.N.

## Author contributions

F.F., M.G., and G.L. designed experiments, analyzed the data, and wrote the manuscript. F.F., W.X., J.-N.H., P.B., T.E., K.W., and R.W. performed in vitro experiments. A.M.S. performed Oncoscan measurements. P.V., S.E., and C.M. designed the shRNA screen. I.A., W.H., G.T., and G.O. performed the histological review and immunohistochemical staining of PBL samples. H.H. performed and evaluated the FISH analysis. I.B., V.B., and P.S. performed targeted resequencing. G.O., J.-T.N., W.H., J.R.G., M.J.B., G.T., J.L.M., I.A., W.K., J.R., F.Fe., A.T., S.D., H.H., and A.R. selected and provided primary PBL samples and analyzed data. F.F., R.W., M.L., M.R.W., E.G.-B., F.C., A.S., J.C., P.A., J.M., T.A., and M.La. acquired clinical data and analyzed data. M.G., B.C., P.L., and M.Z. performed statistical and bioinformatical analysis. All authors read and approved the final manuscript.

## Funding

## Competing interests

The authors declare no competing interests. G.L. received research funding from AstraZeneca.

## Additional information

[1]Department of Medicine A, Department of Hematology, Oncology and Pneumology, University Hospital Münster, Münster, Germany. [2]Dr. Margarete Fischer-Bosch Institute of Clinical Pharmacology, Stuttgart and University of Tuebingen, Tübingen, Germany. [3]Department of Clinical Pathology, Robert Bosch Hospital, Stuttgart, Germany. [4]Institute of Pathology and Neuropathology, Eberhard Karls University of Tübingen-Comprehensive Cancer Center, University Hospital Tübingen, Tübingen, Germany. [5]Department of Hematology, ICO-Hospital Germans Trias i Pujol, Josep Carreras Leukaemia Research Institute (IJC), Universitat Autònoma de Barcelona, Badalona, Spain. [6]Department of Psychiatry and Psychotherapy, University Hospital Freiburg, Freiburg, Germany. [7]Department of Medicine I, Medical Center, Faculty of Medicine, University of Freiburg, Freiburg, Germany. [8]Department of Hematology and Medical Oncology, University Medical Center Göttingen, Göttingen, Germany. [9]Department of Pathology, Queen Elizabeth University Hospital, Glasgow, United Kingdom. [10]Department of Haematology, Beatson West of Scotland Cancer Centre, Glasgow, United Kingdom. [11]Institute of Pathology, University of Würzburg, Würzburg, Germany. [12]Charité Comprehensive Cancer Center, Charité—Universitätsmedizin Berlin, Corporate Member of Freie Universität Berlin, Humboldt-Universität zu

Berlin and Berlin Institute of Health, Berlin, Germany. [13]Department of Hematology. ICO-Hospital Duran i Reynals, Universitat de Barcelona, L'Hospitalet de Llobregat, Llobregat, Spain. [14]Department of Pathology. Hospital Universitari de Bellvitge-IDIBELL, L'Hospitalet de Llobregat, Llobregat, Spain. [15]Department of Hematology, Hospital del Mar, Barcelona, Spain. [16]Department of Pathology, Hospital Universitari Vall d'Hebron, Barcelona, Spain. [17]Department of Hematology, Hospital Universitari Vall d'Hebron, Barcelona, Spain. [18]Department of Pathology, Hospital Gregorio Marañón, Madrid, Spain. [19]Department of Infectious Diseases, Hospital Gregorio Marañón, Madrid, Spain. [20]Division of Hematophathology, Christian-Albrechts-University, Kiel, Germany. [21]Institute of Medical Genetics and Pathology, University Hospital Basel, Basel, Switzerland. [22]Department of Pathology, Hospital Germans Trias i Pujol, Institut Germans Trias i Pujol (IGTP), Universitat Autònoma de Barcelona, Badalona, Spain. [23]Department of Physics, University of Marburg, Marburg, Germany. [24]German Cancer Consortium (DKTK), Partner Site Freiburg, Freiburg, Germany. [25]German Cancer Research Center (DKFZ), Heidelberg, Germany. [26]Division of Translational Pathology, Gerhard-Domagk-Institute of Pathology, University Hospital Münster, Münster, Germany. [27]These authors contributed equally: Georg Lenz, Michael Grau. ✉email: Georg.Lenz@ukmuenster.de

