## [Peer Review File · Nature Communications]

Molecular and functional profiling identifies therapeutically targetable vulnerabilities in plasmablastic lymphomaREVIEWER COMMENTS

Reviewer #1 (Remarks to the Author): Expert in lymphoma, PBL, and therapy

Plasmablastic lymphoma (PBL) a relatively rare and heterogeneous group of lymphomas is characterized by plasmacytic differentiation, frequent association with Epstein Barr virus (EBV) and MYC translocation.

The authors carried out a comprehensive genomic analysis of PBL in a cohort of 99 primary PBL samples. They found alterations activating the RAS-RAF, JAK-STAT and NOTCH pathways, as well as high level amplifications in MCL1 and IRF4 genes. IRF4 and JAK-STAT pathways were identified as promising molecular targets for treatment.

This is the first comprehensive integrated genomic analysis of PBL samples using whole exome sequencing and genome-wide copy number.

The techniques used are reliable as well as the interpretation of data. The conclusions are correctly based on the reported results.

The methodology is sound. The work meets the expected standard in the field; so, the work may be reproduced.

Critical points

In this Reviewer's opinion this is a very demanding job involving a heterogeneous group of lymphomas.

1. The paper is mostly descriptive and lists numerous data obtained with current sophisticated technologies. However, comparative discussion for groups of cases related to EBV, HIV, or neither should be thorough. The spectrum of PBL in fact includes* a) HIV-unrelated PBL, EBV-positive or EBV-negative; b) HIV-related PBL of the oral cavity-type, usually EBV positive; and c) HIV-related PBL usually EBV positive and HHV8 positive which actually corresponds to solid PEL (out of the study). The present study of Frontzek and Colleagues includes patients with HIV-unrelated PBL and HIV-related PBL, either EBV-associated or non-EBV-associated.

2. It would also be very interesting to know if there are morphological differences between the EBV infected and non-infected forms, whatever the epidemiological setting.

*Carbone a et al. Am J Surg Pathol. 2004;28(11):1538-40

Bouvard V et al, Lancet Oncol. 2009 Apr;10(4):321-2.

Carbone A et al. Blood. 2009;113(6):1213-24

Reviewer #2 (Remarks to the Author): Expert in lymphoma genetics and WES

This manuscript reports a very comprehensive molecular and genetic description of plasmablastic lymphoma. Functional genomic studies implicate IRF4 and STAT as potential vulnerabilities of this tumor and point to lenalidomide and interference with cytokine signaling at the IL6R and JAK levels as therapeutic opportunities.

Major comments

1. The distinction of plasmablastic lymphoma and plasma cell neoplasm can be challenging especially for those that are EBV negative. The authors should provide details on how this distinction was made. A central pathology review is strongly advised.

2. The novelty of the manuscript is in the comprehensiveness of the genomic profiling, while the mutational landscape here described is, at least in part, confirmatory (Garcia Reyer et al, *Haematologica*, 2020, Liu et al, *Blood Cancer Discovery* 2020). Leveraging the multilayer dataset here available, the authors should attempt at segregating genetically driven subtypes/clusters, and correlating them with viral status and outcome. I understand the power limitations imposed by the sample size. However, gross subgroups, if any, might be still envisaged.

3. RAS-RAF pathway mutations as well as MYC abnormalities, but not STAT pathway mutations, are also features of plasmablastic myeloma. It would be of interest if the authors could specifically compare their results to plasmablastic myelomas since those are the most likely entity to be confused with plasmablastic lymphomas. Identification of a genetic signature that differentiates plasmablastic lymphoma from plasmablastic myeloma can increase the clinical utility and diagnostic impact of the study.

Specific comments

1. The authors state that: "As expected, 87% (60/69) of PBLs in this series did not express the B-cell antigen CD20, while the remaining 13% (9/69) showed weak and inconsistent expression" – Plasmablastic Lymphoma phenotype is defined, by WHO consensus as a terminal B cell phenotype with loss of B cell markers and variable acquisition of plasma cell differentiation markers in a proliferating cell. The authors should specify whether all cases underwent central pathology review by independent pathologists. Cases showing CD20 expression should be carefully reassessed. The phenotype and EBV status of cases (n=30) that were not assembled in TMA should also be reported.

2. The authors state that: "To characterize the mutational landscape of PBL, we performed WES in 88 primary PBL cases" – The authors should clarify why 11 cases were not tested by WES. More in general, a CONSORT diagram detailing the number of cases out of the total that have been assessed by each technology (eg, TMA, FISH, WES, Oncoscan...) and a justification for the missing cases should be provided.

3. The authors state that: "...we examined lymphoma specific survival (LSS) in the cohort of patients treated with CHOP-like chemotherapy, in order to exclude potential bias of disease-unrelated death causes and inefficient treatment approaches" – The authors should rephrase the statement. Indeed, LLS adjust solely for lymphoma unrelated deaths.

Response to reviewer 1:

1. The paper is mostly descriptive and lists numerous data obtained with current sophisticated technologies. However, comparative discussion for groups of cases related to EBV, HIV, or neither should be thorough. The spectrum of PBL in fact includes a) HIV-unrelated PBL, EBV-positive or EBV-negative; b) HIV-related PBL of the oral cavity-type, usually EBV positive; and c) HIV-related PBL usually EBV positive and HHV8 positive which actually corresponds to solid PEL (out of the study). The present study of Frontzek and Colleagues includes patients with HIV-unrelated PBL and HIV-related PBL, either EBV-associated or non-EBV-associated.

We agree with the reviewer to investigate the potential impact of Epstein-Barr virus (EBV) and human immunodeficiency virus (HIV) infection, two hallmarks in the pathogenesis and classification of PBL, on the incidence of genetic aberrations. To this end, we have determined the EBV status of all included plasmablastic lymphoma (PBL) cases to increase the power of our analyses. All HHV8 positive cases were excluded (revised Supplementary Figure 1a). Next, we have analyzed the distribution of recurrent mutations (frequency $\geq 5\%$, q-value ≤ 0.1 and mutations of *MYC*, *BRAF* as shown in Figure 1a) and somatic copy number alterations (SCNAs) (frequency $\geq 5\%$, q-value ≤ 0.1) with respect to the EBV and HIV infection status. More precisely, we have compared HIV+ (n=17) vs. HIV- (n=35), EBV+ (n=55) vs. EBV- (n=41), HIV+/EBV+ (n=15) vs. HIV+/EBV- (n=2), and HIV-/EBV+ (n=17) vs. HIV-/EBV- (n=18) PBL utilizing one-tailed Wilcoxon rank sum tests on available data for each selected genetic lesion as described in the revised methods on page 18 (revised Supplementary Table 6). In HIV-infected individuals, we detected *STAT3* mutations significantly more frequently compared to HIV negative patients (47% vs. 10%, p=0.0025, q=0.0429) while recurrent amplifications and deletions did not significantly differ as described on pages 4-5 and on page 8 of the revised manuscript. In patients with EBV negative PBL, we detected focal deletions of 1p22.1 (46% vs. 11%, p=0.0007, q=0.0138) and arm-level deletions of 13q (27% vs. 7%, p=0.0045, q=0.0423) as characteristic genetic alterations in comparison to EBV positive disease (page 8 of the revised manuscript). In contrast, we did not detect any significant differences with respect to recurrent mutations although we noticed a non-significant trend for accumulation of *TP53* mutations in EBV negative cases (24% vs 6%, p=0.0114, q=0.1938; page 5 of the revised manuscript). Possibly in part due to the limited number of cases, no significant differences could be detected in the subgroup comparisons of HIV-/EBV+ vs. HIV-/EBV- and HIV+/EBV+ vs. HIV+/EBV- PBL. These results were outlined on page 8 of the revised manuscript.

As stated by the reviewer, one typical feature of PBL represents its frequent involvement of the oral cavity (Delecluse et al., 1997). In our cohort, we had respective clinical

data for 54 patients. While we identified 18 PBL cases with involvement of the oral cavity, in 36 patients the oral cavity was not affected (revised Supplementary Table 1). Interestingly, mutations of *CFAP44* occurred significantly more frequently in PBL arising in the oral cavity (27% vs. 0%, $p=0.0016$, $q=0.0268$). We described this novel finding on page 8 of the revised manuscript. In contrast, the distribution of SCNAs did not significantly differ.

2. It would also be very interesting to know if there are morphological differences between the EBV infected and non-infected forms, whatever the epidemiological setting.

The reviewer raises an important point. Thus, we have thoroughly re-reviewed cases and analyzed the distribution of the following morphologic variants between EBV positive and EBV negative PBL: typical monomorphic morphology (immunoblastic/plasmablastic) versus presence of plasmacytoid differentiation, presence of anaplastic/giant cells, presence of necrosis, and presence of a starry sky pattern. However, these analyses did not detect any significant associations of EBV status and morphologic variants. These analyses and results were summarized in the revised manuscript on page 3.

Response to reviewer 2:

1. The distinction of plasmablastic lymphoma and plasma cell neoplasm can be challenging especially for those that are EBV negative. The authors should provide details on how this distinction was made. A central pathology review is strongly advised.

We completely agree with the reviewer. All PBL cases enrolled in our study were contributed by expert hematopathological institutions. Due to the challenging diagnosis of PBL, selected primary PBL cases were independently reevaluated in a central pathology review of 4 expert hematopathologists (Ioannis Anagnostopoulos, Wolfgang Hartmann, German Ott, and Gustavo Tapia). Individual cases that were subsequently included in our analysis were evaluated by at least two different expert hematopathologists. In general, we diagnosed PBL according to the criteria of the updated WHO classification of 2017. Therefore, all cases displayed an immunoblastic and/or plasmablastic morphology with a minority of them showing morphological plasmacytoid differentiation.

We agree with reviewer 2 that the distinction of plasmablastic lymphoma and plasmablastic plasma cell myeloma can be challenging or in some cases even impossible on morphological and immunophenotypic basis alone. According to our diagnostic approach, all cases were required to have plasmablastic morphology, thus primarily excluding, for example, extramedullary plasmacytoma typically arising in the upper aerodigestive tract. Furthermore, all cases with prior or concomitant diagnosis of a plasma cell neoplasm as well as all tumors only involving bone structures were excluded from our study, thus leaving a series of *bona fide* PBLs. ALK+ DLBCL and HHV8-associated lymphoproliferations were excluded based on respective immunohistochemical stainings.

Overall, 22 cases that did not fulfill the diagnostic criteria of PBL were excluded from our study cohort, as indicated in the revised Supplementary Figure 1a. Twelve cases were classified as plasma cell neoplasms. This algorithm was described in the revised manuscript on page 14. Of the 22 excluded cases, three were excluded during the re-analysis in this rebuttal phase. Thus, we recalculated all analyses for the revised manuscript.

2. The novelty of the manuscript is in the comprehensiveness of the genomic profiling, while the mutational landscape here described is, at least in part, confirmatory (Garcia Reyero et al, Haematologica, 2020, Liu et al, Blood Cancer Discovery 2020). Leveraging the multilayer dataset here available, the authors should attempt at segregating genetically driven subtypes/clusters, and correlating them with viral status and outcome. I understand the power limitations imposed by the sample size. However, gross subgroups, if any, might be still envisaged.

We thank the reviewer for raising this important point. Accordingly, we first applied unsupervised hierarchical clustering to all significantly mutated genes and somatic copy number alterations (SCNAs) (frequency ≥ 0.05 , $q \leq 0.1$) in our study, including all samples with both whole exome sequencing (WES) and Oncoscan data available (n=73). To identify correlated genomic features, we selected the correlation distance metric and used average linkage. The resulting clustering plot is shown in Rebuttal Figure 1. While one single cluster of features was detected, further analyses revealed that this cluster represents a trivial clustering of broad and focal lesions located on the chromosomal arm 1q. Unfortunately, no additional significant clusters were detected using this unsupervised approach. As these data are completely negative in nature, we decided not to include these data in the revised manuscript. However, to further leverage our multi-omics dataset as suggested, we performed supervised statistical comparisons of significant genetic alterations in biologically defined PBL subcohorts. More precisely, we have compared the distribution of recurrent significant mutations and SCNAs in HIV positive (n=17) vs. HIV negative patients (n=35), EBV positive (n=55) vs. EBV negative disease (n=41), HIV positive patients with EBV positive PBL (HIV+/EBV+, n=15) vs. HIV positive patients with EBV negative PBL (HIV+/EBV-, n=2), HIV negative patients with EBV positive PBL (HIV-/EBV+, n=17) vs. HIV negative patients with EBV negative PBL (HIV-/EBV-, n=18), *MYC* translocated (n=28) vs. *MYC* untranslocated PBL (n=32), CD20 negative (n=82) vs. weakly CD20 positive PBL (n=13), PBL arising in the oral cavity/pharynx (n=18) vs. PBL arising elsewhere (n=36), patients with (n=32) vs. without immunosuppression (n=29), patients with high IPI (n=9) vs. low/intermediate IPI (n=32), patients with lymphoma-specific survival (LSS) of less than 12 months (n=8) vs. patients with LSS of more than 24 months (n=15) (revised Supplementary Table 6). We utilized one-tailed Wilcoxon rank sum tests on available data for each selected genetic lesion as described in our revised methods on page 18.

First, we analyzed the impact of viral infection status on genetic heterogeneity. In HIV-infected individuals, we detected *STAT3* mutations significantly more frequently compared to HIV negative patients (47% vs. 10%, $p=0.0025$, $q=0.0429$) while recurrent amplifications and deletions did not significantly differ. These results were stated on pages 4-5 and on page 8 of the revised manuscript. In patients with EBV negative PBL, we detected focal deletions of 1p22.1 (46% vs. 11%, $p=0.0007$, $q=0.0138$) and arm-level deletions of 13q (27% vs. 7%, $p=0.0045$, $q=0.0423$) as characteristic genetic alterations in comparison to EBV positive PBL. These results were summarized on page 8 of the revised manuscript. We did not detect any significant differences in the pattern of recurrent mutations although we noticed a non-significant trend for accumulation of *TP53* mutations in EBV negative PBL (24% vs 6%, $p=0.0114$, $q=0.1938$; stated on page 5 of the revised manuscript). Possibly in part due to limited case numbers, no significant differences were detectable in the subgroup comparisons of HIV-/EBV+ vs. HIV-/EBV- and HIV+/EBV+ vs. HIV+/EBV- (page 8).

One genetic hallmark of PBL represents frequent translocations of *MYC*. In the subgroup of PBLs harboring a *MYC* translocation, we detected a pattern of several focal amplifications that were significantly more frequent compared to PBLs without *MYC* translocation: 1q43 (58% vs. 21%, $p=0.0122$, $q=0.0754$), 2q31.3 (46% vs. 14%, $p=0.0019$, $q=0.0603$), 11q23.3 (50% vs. 18%, $p=0.0046$, $q=0.0619$), 11q25 (46% vs. 14%, $p=0.0122$, $q=0.0754$), 12p11.22 (42% vs. 14%, $p=0.0060$, $q=0.0619$). These new results are described on page 8 of the revised manuscript. Recurrent deletions did not differ. Comparing recurrent mutations, we found that *MYC* mutations only occurred in PBLs with concomitant *MYC* translocation (27% vs 0%, $p=0.0010$, $q=0.0171$) as stated on page 6 of the revised manuscript.

Classically, PBLs do not express classical B-cell markers. However, a minority of PBLs still displays a weak CD20 expression, 14% (13/95) of cases in our cohort (also see point 4 of reviewer 2). Comparing the pattern of mutations and SCNAs of CD20 negative and weakly CD20 positive PBL cases, we did not find any significant differences. These results were stated on page 8 of the revised manuscript.

Finally, we also included our clinical data in this analysis. As initially described by Delecluse et al., PBL frequently arises in the oral cavity. In our cohort, we had respective clinical data for 54 patients. While we identified 18 PBL cases with involvement of the oral cavity in 36 patients the oral cavity was not affected (revised Supplementary Table 1). Interestingly, mutations of *CFAP44* occurred significantly more often in PBL arising in the oral cavity (27% vs. 0%, $p=0,0016$, $q=0,0268$). We described this novel finding on page 8 of the revised manuscript. The distribution of SCNAs did not significantly differ. Apart from HIV infection, PBL patients often suffer from immunodeficiency caused by autoimmune diseases, organ transplantation or other viral infections such as chronic hepatitis C infection (reported in the revised Supplementary Table 1). In our cohort, we detected focal deletions of 4q35.2 (50% vs. 11%, $p=0.0035$, $q=0.0671$) and broad deletions of 18p (33% vs. 4%, $p=0.0083$, $q=0.0788$) significantly more frequent in immunocompetent compared to immunocompromised patients. These results were described on page 8 of the revised manuscript. Recurrent amplifications or mutations did not significantly differ. Possibly also due to limited case numbers, we did not reveal any significant differences comparing PBL patients with high IPI vs. patients with low/intermediate IPI as well as comparing patients with lymphoma-specific survival of less than 12 months vs. patients with lymphoma-specific survival of more than 24 months (page 8 of the revised manuscript).

3. RAS-RAF pathway mutations as well as MYC abnormalities, but not STAT pathway mutations, are also features of plasmablastic myeloma. It would be of interest if the authors could specifically compare their results to plasmablastic myelomas since those are the most likely entity to be confused with plasmablastic lymphomas. Identification of a genetic signature

that differentiates plasmablastic lymphoma from plasmablastic myeloma can increase the clinical utility and diagnostic impact of the study.

As outlined above in detail, plasmablastic myeloma represents a difficult differential diagnosis of PBL. For borderline cases, a definite diagnosis cannot solely be made based on histological and immunohistochemical criteria. In these cases, clinical parameters have to be considered and finally guide diagnosis and subsequent treatment. Presence of CRAB criteria favors the diagnosis of plasmablastic myeloma while e.g. EBV and/or HIV infection are typical features of PBL. In our PBL cohort, two thirds of cases were EBV positive and one third of patients was infected by HIV. Available clinical information were compatible with the diagnosis of PBL (Supplementary Table 9). All cases with previous or concomitant history of plasma cell neoplasm were excluded. Thus, we believe that our cohort truly consists of *bona fide* PBL cases.

While we agree with reviewer 2 that a genetic signature differentiating PBL from plasmablastic myeloma would be desirable, its generation is unfortunately out of the scope of the current manuscript, as we would need to acquire a novel cohort of clear plasmablastic myeloma cases that we would need to analyze by whole exome sequencing and Oncoscan. A very small study by Liu et al. (The American Journal of Surgical Pathology, 2020) describes a case series of ten plasmablastic myelomas. None of these cases exhibited tumor cells latently infected by EBV, and no alterations affecting genes encoding members of the JAK-STAT pathway were reported in a targeted sequencing analysis. However, significantly larger and more comprehensive analyses are warranted to identify the molecular differences and similarities between PBL and plasmablastic myeloma. We integrated this important point in our discussion on page 12 of the revised manuscript.

4. The authors state that: "As expected, 87% (60/69) of PBLs in this series did not express the B-cell antigen CD20, while the remaining 13% (9/69) showed weak and inconsistent expression" – Plasmablastic Lymphoma phenotype is defined, by WHO consensus as a terminal B cell phenotype with loss of B cell markers and variable acquisition of plasma cell differentiation markers in a proliferating cell. The authors should specify whether all cases underwent central pathology review by independent pathologists. Cases showing CD20 expression should be carefully reassessed. The phenotype and EBV status of cases (n=30) that were not assembled in TMA should also be reported.

As described above, due to the challenging diagnosis of PBL, selected primary PBL cases were independently reevaluated in a central pathology review of 4 expert hematopathologists (Ioannis Anagnostopoulos, Wolfgang Hartmann, German Ott, and Gustavo Tapia). Individual

cases that were subsequently included in our analysis were evaluated by at least two different expert hematopathologists. This central pathology review is outlined in the methods on page 14 of the revised manuscript. According to the updated WHO classification of 2017, B-cell markers such as CD45, CD20, and PAX5 are either negative or sometimes weakly positive in a minority of PBL cells so that weak positivity for CD20 does not represent an exclusion criterion. However, we reassessed CD20 immunophenotype carefully. Overall, 13 of 95 PBL (14%) cases showed patchy, inconsistent, and generally weak CD20 expression as described on page 3 of the revised manuscript. “CD20 positive” PBL specimens never featured strong or uniform reactivity of tumor cells setting all cases clearly apart from CD20 positive lymphomas such as diffuse large B-cell lymphoma (DLBCL). Finally, we compared the occurrence of recurrent significant mutations and SCNAs in classically CD20 negative versus weakly CD20 positive PBLs. These analyses did not reveal any significant differences (page 8 of the revised manuscript, revised Supplementary Table 6).

Additionally, we determined the EBV infection status of all included cases as requested. Overall, 57% (55/96) of PBL cases exhibited latently EBV-infected tumor cells (page 3, revised Suppl. Fig 1c and Suppl. Table 1). All depending analyses were correspondingly reperformed.

5. The authors state that: “To characterize the mutational landscape of PBL, we performed WES in 88 primary PBL cases” – The authors should clarify why 11 cases were not tested by WES. More in general, a CONSORT diagram detailing the number of cases out of the total that have been assessed by each technology (eg, TMA, FISH, WES, Oncoscan...) and a justification for the missing cases should be provided.

We agree and thank the reviewer for this important suggestion. Accordingly, we have added a CONSORT diagram indicating the number of included cases and all reasons for exclusion. The CONSORT diagram is shown in the revised Supplementary Figure 1a.

6. The authors state that: “...we examined lymphoma specific survival (LSS) in the cohort of patients treated with CHOP-like chemotherapy, in order to exclude potential bias of disease-unrelated death causes and inefficient treatment approaches” – The authors should rephrase the statement. Indeed, LLS adjust solely for lymphoma unrelated deaths.

We agree with the reviewer and we have rephrased the statement on page 7 of the revised manuscript.

Frontzek et al. Rebuttal Figure 1

Rebuttal Figure 1: Unsupervised clustering.

Unsupervised hierarchical clustering of all significantly mutated genes and SCNAs ($q \leq 0.1$, cohort frequency ≥ 0.05) for all samples and the cell line PBL-1 with both WES and Oncoscan data available (n=73). Correlation distance metric and average linkage were used for both, feature and sample dendrograms. Cluster A represents a trivial co-clustering of increased copy numbers in broad and focal SCNA segments located on the chromosomal arm 1q.

REVIEWER COMMENTS

Reviewer #1 (Remarks to the Author):

My concerns have been addressed in the revised manuscript.

Reviewer #2 (Remarks to the Author):

The authors addressed all issues raised by the reviewers

Reviewer #4 (Remarks to the Author):

Frontzek et al. present a comprehensive molecular characterization of a large (approx. n=90) cohort of plasmablastic lymphoma combined with functional analyses including a shRNA screen and in vitro cell line experiments to identify dependencies on IRF4 and JAK-STAT3 signaling. Overall, this is a comprehensive and informative analysis of this rare and difficult to diagnose disease entity. This review explicitly focusses on the technical aspects of molecular analyses, since other aspects have already been reviewed and considerably improved within a prior round of revision.

The two main technical problems of this study were that the whole exome sequencing and SNP array analyses had to be carried out from DNA isolated from FFPE Material and that not all samples had available germline DNA samples. The first aspect confounds quality of the data on the level of DNA integrity, the second aspect makes precise calling of somatically acquired variants more difficult. Therefore, this data will always be a tradeoff between reduced input sample quality, difficult bioinformatics and the highly warranted need to investigate these rare clinical samples at all.

In their study, the investigators have already partly openly acknowledged these two drawbacks and addressed these problems by adapted use of the appropriate kits and protocols for library preparation and design of an elaborately customized bioinformatic approach, in order to best as possibly compensate for these technical obstacles.

While the authors provide extremely elaborate data in supplementary tables, it is not feasible for a reviewer to check all of this data by hand. The manuscript would therefore greatly benefit from an additionally more structured and summarized handling of the primary molecular data addressing the following points:

Major:

It is known that DNA quality from FFPE samples is highly dependent on the age of the FFPE samples. Can the authors provide data on the age of the investigated samples and possible correlation with DNA QC parameters going beyond QUBIT measurement, - e.g. Tape station / gel-electrophoresis or other? How does this correlate with total mutational burden (TMB) of the individual samples after final mutational calling?

In such a paper, it would generally be good to see exemplary primary data of the molecular analyses in addition to the aggregated and summarized data. i.e. show examples of the MYC FISH data, show informative exemplary single chromosome data of the SNP array

analysis (there would be plenty of space for this in Figure 2) – e.g. show some examples of del17p or the difficult to analyse IRF4 locus in a single chromosome view.

Page 14: “The median tumor content was 90% (min-max range: 20-100%)”: How was tumor content determined? How does tumor content data correlate with TMB with the applied calling strategy?

Not a single mutation was validated by an independent sequencing method. I would strongly recommend to perform some validation for at least a subset of significantly called mutations. Could hotspot variants e.g. such as KRAS / STAT3 / TP53 have been missed due to low coverage? Therefore screening the whole cohort with a more sensitive and quantitative targeted assay might increase the frequency of mutations and accuracy of CCF and the derived conclusions.

What kind of samples were used as germline controls when available from patient samples and can the authors indicate levels of tumors contamination therein?

Could the authors provide an overview about the landscape / TMB of mutational cohorts for the subcohort with and without matched germline controls?

Response to reviewer #4:

1. While the authors provide extremely elaborate data in supplementary tables, it is not feasible for a reviewer to check all of this data by hand. The manuscript would therefore greatly benefit from an additionally more structured and summarized handling of the primary molecular data (...).

We agree with the reviewer. To address this point, we depict our analysis pipeline in the new Supplementary Fig. 3 summarizing the data flow, the analysis structure, and the resource databases that we have used for our analyses.

2. It is known that DNA quality from FFPE samples is highly dependent on the age of the FFPE samples. Can the authors provide data on the age of the investigated samples and possible correlation with DNA QC parameters going beyond QUBIT measurement, - e.g. Tape station / gel-electrophoresis or other? How does this correlate with total mutational burden (TMB) of the individual samples after final mutational calling?

We agree that formaldehyde fixation and tissue dehydration lead to degradation of DNA representing a technical challenge for further analyses. To assess the quality of FFPE DNA, we applied a qPCR based quality control method by Illumina (Illumina FFPE QC kit) comparing the amplifiability of the extracted DNA to a reference template. Obtained ΔC_t values (cycle threshold) indicated good DNA quality when a value of ≤ 2 was determined (as stated in the revised manuscript on page 15). For all 85 primary PBL samples included in our WES analysis, we determined corresponding ΔC_t values as well as the age of the FFPE specimens. We correlated both parameters and found a significant correlation: higher age correlated to decreased DNA quality expressed by higher ΔC_t values ($r=0.53$, $p=1.5 \cdot 10^{-7}$; new Supplementary Fig. 11a). Further, we correlated the number of called somatic mutations to age of the sample without finding a significant relation ($r=-0.2$, $p=0.07$). We next associated the number of mutations with measured ΔC_t values and did not detect a significant correlation either ($r=-0.02$, $p=0.84$), suggesting that decreased quality of extracted FFPE DNA does not affect the total mutational burden (TMB; new Supplementary Fig. 11b). This underlines that our applied filtering strategy is effective at recognizing FFPE-specific artefacts and compensates for less quality of the utilized DNA. These new results and quality controls are described on pages 16-17 of the revised manuscript.

3. In such a paper, it would generally be good to see exemplary primary data of the molecular analyses in addition to the aggregated and summarized data. i.e. show examples of the MYC

FISH data, show informative exemplary single chromosome data of the SNP array analysis (there would be plenty of space for this in Figure 2) – e.g. show some examples of del17p or the difficult to analyse IRF4 locus in a single chromosome view

We thank the reviewer for this valuable comment. Accordingly, we integrated primary data of our molecular analyses in the manuscript for further illustration. The new Supplementary Fig. 2a shows a representative example of a *MYC* translocation in a PBL case using a *MYC* dual-color break apart probe. In case of a *MYC* translocation, the normally co-localized red and green fluorescence signals get separated in at least 15% of analyzed cells. Applying a *MYC-IgH* fusion probe allows detection of a *MYC-IgH* translocation. Red and green fluorescence signals mark the gene loci of *MYC* and *IgH*, respectively. In case of a *MYC-IgH* translocation, the signals fuse in at least 9% of analyzed cells as representatively shown in Supplementary Fig 2b.

One key finding of our SNP array analysis represents the recurrent focal amplification of the gene locus 6p25.3 including the gene encoding the transcription factor *IRF4*. To illustrate this specific finding, we have integrated a pileup plot showing the number of PBL samples with amplifications of 6p25.3 in our revised Figure 2b. The minimal common region (MCR) determined by GISTIC and our extended peak region (as described in detail in our methods section on page 19) are visualized in green color. Contained genes are labelled. The MCR starts with the first available SNP probe (marked by red crosses) on this chromosomal arm while only non-coding genes *LOC285766* and *LINC00266-3* are located left in the extrapolated extended peak region (striped). These results are described on pages 7 and 19 of the revised manuscript.

4. Page 14: “The median tumor content was 90% (min-max range: 20-100%)”: How was tumor content determined? How does tumor content data correlate with TMB with the applied calling strategy?

The tumor content was microscopically determined on hematoxylin and eosin stained slides by our expert hematopathologists. Specifically, the relative percentage of tumor blastic B-cells was recorded in increments of 10% taking into account the relative percentages of non-blastic bystander lymphocytes and histiocytes (page 15 of the revised methods). As described the median tumor content was 90% and in total only 8 samples displayed a tumor content below 70%. For all samples with available data, we correlated the estimated tumor content to the number of called mutations per sample without detecting a significant association ($r=0.04$, $p=0.75$; new Supplementary Fig. 11c). The few samples with lower tumor content did not show decreased TMB indicating sufficient material for mutation calling.

5. Not a single mutation was validated by an independent sequencing method. I would strongly recommend to perform some validation for at least a subset of significantly called mutations. Could hotspot variants e.g. such as *KRAS* / *STAT3* / *TP53* have been missed due to low coverage? Therefore screening the whole cohort with a more sensitive and quantitative targeted assay might increase the frequency of mutations and accuracy of CCF and the derived conclusions.

We agree with the reviewer. As requested, we have now validated identified hotspot mutations of selected cancer candidate genes (CCGs) in 54 primary PBL samples for which sufficient DNA was still available for independent resequencing. We applied deep targeted resequencing (TRG) using an AmpliSeq custom panel covering the hotspot regions of the selected CCGs *NRAS*, *KRAS*, *BRAF*, *STAT3*, and the complete coding sequences of *TP53* and *TET2*. Amplicon library preparation and sequencing were performed according to manufacturers' instructions. Variant calling was computed using the Ion Reporter Software (this is described in detail in the revised methods on page 18).

Next, we systematically compared called mutations by WES and TRG in all regions of selected CCGs covered by amplicons of the targeted library. All variants identified by TRG are listed in the new Supplementary Table 7. Overall, of 79 variants identified in the targeted resequencing approach, 48 mutations were identically called by both technologies. Only one single *KRAS* mutation called by WES (p.G12D) could not be validated in TRG although this variant was confirmed in WES reads using IGV (of 94 total reads, 59 showed reference base C and 35 showed mutated base T). For nine cases, the wildtype status determined by WES was confirmed.

As expected, deep TRG revealed further variants that were either not detected in WES or filtered out for specific reasons. Specifically, TRG revealed nine variants with low allele frequency (AF; <10%). This is below the defined threshold of our WES analysis as we intentionally focused on subclones of major biological significance. Consistently, we used an AF \geq 10% as one filter step for variant calling in WES. One additional variant with an AF of 13% in TRG had AF of <10% in WES and was consequently filtered out. TRG identified 11 further variants that were equally detected by WES but subsequently filtered out as germline or due to an artefact. TRG revealed six new variants of higher allele frequency that were not called by WES due to low effective coverage (less than 20 reads). Additional three variants were not called by a combination of moderate coverage and relatively low AF (10-16%). These nine variants detected in six different PBL cases are: *STAT3*: PBL006 (c.1981G>T), PBL114 (c.1842C>A and c.1852G>C), PBL037(c.1842C>G); *TP53*: PBL009 (c.818G>T and c.827C>A), PBL039 (c.493C>T), PBL058 (c.326T>G); and *KRAS*: PBL037 (c.38G>A). This

analysis suggests that individual hotspot variants might be missed due to lower coverage in WES for single samples and genes. Specifically, *STAT3* mutation frequency slightly increased from 25% (21/85) to 28% (24/85), for *TP53* from 14% (12/85) to 18% (15/85), and for *KRAS* from 11% (9/85) to 12% (10/85). Interestingly, no further *NRAS*, *BRAF*, or *TET2* mutations were identified by TRG although *NRAS* depicts the highest cohort mutation frequency of 31% in WES. This indicates a good coverage for these genes over all samples. Furthermore, we already had performed a coverage QC showing that most genes were covered with sufficient reads in most samples (Supplementary Figure 4). Overall, only 5.2% of covered genomic regions had an effective coverage of less than 20 reads.

To summarize, TRG validation confirmed all identified hotspot mutations in the selected CCGs except of one single *KRAS* mutation that is clearly present in the WES reads. Further mutations discovered by TRG were filtered as germline or showed low AF below the defined threshold of our WES study. Only a small number of new detected variants has been missed due to low effective coverage in our WES approach only marginally increasing the mutational frequencies. These new results are integrated and discussed in detail in the revised manuscript on page 6.

6. What kind of samples were used as germline controls when available from patient samples and can the authors indicate levels of tumors contamination therein?

We thank the reviewer for this comment. For all used germline controls, we included a list of the sites of origin in the revised Supplementary Table 1. All used normal controls were carefully examined by our expert hematopathologists. For all cases, contamination with tumor cells was excluded based on conventional hematoxylin/ eosin staining and immunohistochemistry.

7. Could the authors provide an overview about the landscape / TMB of mutational cohorts for the subcohort with and without matched germline controls?

We thank the reviewer for raising this important point. Indeed, matched germline samples generally allow for more specific variant filtering by Mutect. Therefore, we have systematically compared the subcohorts of PBL cases with matched normal (n=22) vs. PBL cases without available paired normal tissue (n=65).

As expected, the global TMB measured over the complete WES library was generally lower in tumor cases with corresponding normal tissue. The median TMB per megabase was 1.87 for PBL cases with paired normals vs. 3.58 for PBL cases without (stated on page 6 of the revised manuscript). To further investigate the impact on top mutated genes with a cohort frequency of more than 10%, we compared the distribution of recurrent mutations between

both subcohorts utilizing one-tailed Wilcoxon rank sum tests (revised Supplementary Table 6). Corresponding oncoplots are shown in the new Supplementary Fig. 8 and 9. Adjusting for multiple hypothesis testing, our statistical analysis revealed only *MUC4* as significantly more frequently mutated gene in the cohort of PBL samples without a matched normal control ($p=0.0004$, $q=0.0194$; Supplementary Table 6). However, *MUC4* had already been filtered out at the gene level quality control as it showed artificially high coverage, most likely due to sequence homology (Supplementary Fig. 4). Additionally, various rare SNPs have been described for *MUC4* with population frequencies far less than 5%. These SNPs are not part of the NCBI common variants dataset utilized by the dbSNP filtering step. Importantly, for all other frequently mutated genes we did not reveal any significant differences, i.e. there is no systematic bias in our analysis when determining the mutational frequencies for patient samples without available paired normal tissue. This suggests that our filtering strategy and, in particular, our created panel of normals enables sufficiently specific calling of somatic mutations. These new results are described in detail on page 6 of the revised manuscript.

Once again we would like to thank the reviewers for the evaluation of our manuscript.

REVIEWER COMMENTS

Reviewer #4 (Remarks to the Author):

In their revised version the authors have addressed all issues of the review adequately.